# Boosting Virtual Agent Learning and Reasoning: A Step-Wise, Multi-Dimensional, and Generalist Reward Model with Benchmark

Bingchen Miao [1 2 *]   Yang Wu [2 *]   Minghe Gao [1 *]   Qifan Yu [1]   Wendong Bu [1 2]   Wenqiao Zhang [1]   Yunfei Li [2]   Siliang Tang [1]   Tat-Seng Chua [3]   Juncheng Li [1 ✉]

## Abstract

The development of Generalist Virtual Agents (GVAs) has shown significant promise in autonomous task execution. However, current training paradigms face critical limitations, including reliance on outcome supervision and labor-intensive human annotations. To address these challenges, we propose **Similar**, a step-wise multi-dimensional generalist reward model, which offers fine-grained signals for agent training and can choose better actions for inference-time scaling. Specifically, we begin by systematically defining five dimensions for evaluating agent actions. Building on this framework, we design an MCTS-P algorithm to automatically collect and annotate step-wise, five-dimensional agent execution data. Using this data, we train **Similar** with our crafted Triple-M strategy. Furthermore, we introduce the first benchmark in the virtual agent domain for step-wise, multi-dimensional reward model training and evaluation, named *SRM*. This benchmark consists of two components: *SRMTrain*, which serves as the training set for **Similar**, and *SRMEval*, a manually selected test set for evaluating the reward model. Experimental results demonstrate that **Similar**, through its step-wise, multi-dimensional assessment and synergistic gain, provides GVAs with effective intermediate signals during both training and inference-time scaling. The code is available in https://github.com/antgroup/Similar.

## 1. Introduction

Generalist Virtual Agents (GVAs (Gao et al., 2024; Bu et al., 2025)) powered by Multimodal Large Language Models

(MLLMs (Li et al., 2023a;b; Pan et al., 2025; 2024b)) process multimodal inputs (UI elements (Zhang et al., 2024a), text (Shen et al., 2023), visuals (Yan et al., 2023)) to navigate digital environments, performing tasks and generating outputs that manipulate interfaces or provide responses. The training of GVAs relies on outcome-based rewards from human-annotated trajectories, where task completion serves as the primary supervision signal (He et al., 2024).

However, this paradigm with the outcome reward for GVAs has significant limitations. **1) Lack of multi-dimensional fine-grained process supervision:** Existing methods typically focus on global task success or the final state of the task, overlooking intermediate steps in execution (Yu et al., 2024a). This oversight makes it impossible to pinpoint failures in unsuccessful trajectories or identify errors in successful ones, resulting in inefficient learning and reasoning processes (Uesato et al., 2022; Lightman et al., 2023; Gao et al., 2025). In contrast, a Process Reward Model (PRM) offers a better alternative by providing fine-grained supervision signals to guide agent behavior. **2) Reliance on human-annotated trajectories with reward signals:** Domain experts need to meticulously annotate trajectories consisting of dozens of steps with accurate outcome-based rewards to train GVAs (He et al., 2024). Furthermore, obtaining step-wise fine-grained process-based rewards makes the process labor-intensive, time-consuming, and nearly infeasible at scale (Deng et al., 2023; Burns et al., 2022). **3) Difficulty in scaling inference-time**. Recent outstanding work has demonstrated that inference-time scaling can significantly enhance agent performance (DeepSeek-AI et al., 2025; Wu et al., 2024b). However, relying on result-based training with extensive human annotation limits the ability to handle complex tasks by selecting the best action or step-by-step assessments. (Snell et al., 2024; Zelikman et al., 2022). Therefore, our focus has shifted to breaking result-oriented manual annotation-dependent training methods through step-wise automatic reward model.

To address these challenges, we propose **Similar**, a step-wise **m**ulti-dimensional gener**a**list **r**eward model. It provides fine-grained supervision signals for agent training and inference-time scaling, enabling automated, multi-faceted assessment without relying on labor-intensive human an-

---
*Equal contribution  [1]Zhejiang University, Hangzhou, China [2]Ant Group, Hangzhou, China [3]National University of Singapore, Kent Ridge, Singapore. Correspondence to: Juncheng Li <junchengli@zju.edu.cn>.

*Proceedings of the 42nd International Conference on Machine Learning*, Vancouver, Canada. PMLR 267, 2025. Copyright 2025 by the author(s).

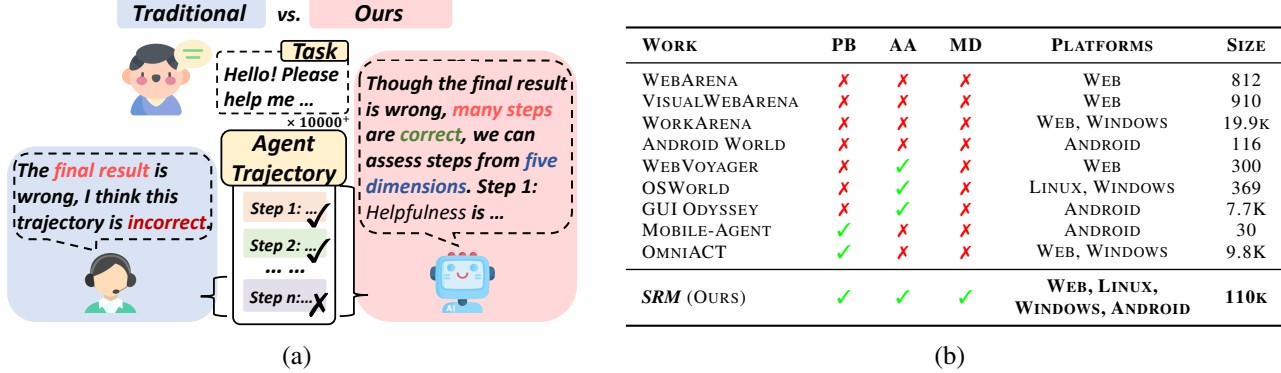

| WORK | PB | AA | MD | PLATFORMS | SIZE |
|------|----|----|----|-----------|------|
| WEBARENA | ✗ | ✗ | ✗ | WEB | 812 |
| VISUALWEBARENA | ✗ | ✗ | ✗ | WEB | 910 |
| WORKARENA | ✗ | ✗ | ✗ | WEB, WINDOWS | 19.9K |
| ANDROID WORLD | ✗ | ✗ | ✗ | ANDROID | 116 |
| WEBVOYAGER | ✗ | ✓ | ✗ | WEB | 300 |
| OSWORLD | ✗ | ✓ | ✗ | LINUX, WINDOWS | 369 |
| GUI ODYSSEY | ✗ | ✓ | ✗ | ANDROID | 7.7K |
| MOBILE-AGENT | ✓ | ✗ | ✗ | ANDROID | 30 |
| OMNIACT | ✓ | ✗ | ✗ | WEB, WINDOWS | 9.8K |
| *SRM* (OURS) | ✓ | ✓ | ✓ | WEB, LINUX, WINDOWS, ANDROID | 110K |

| (a) | (b) |
|-----|-----|

Figure 1: (a) Traditional coarse-grained outcome-based labor-intensive paradigm vs. Our fine-grained process-based autonomous paradigm. (b) Comparison of benchmark platforms. Previous works focused on virtual agent benchmarks, while ours is the first benchmark specifically for virtual agent reward models. The PB, AA, and MD in the list represent Process-based, Automatic Annotation, and Multi-Dimension, respectively.

notations. Specifically, **1)** we introduce a step-wise, multi-dimensional assessment system for GVA actions, **defining five key dimensions of process supervision signals**: *Helpfulness*, *Odds of Success*, *Efficiency*, *Task Relevance*, and *Coherence*. These dimensions are designed to minimize overlap while collectively providing a comprehensive assessment of each action's quality. **2)** Then we design an **MCTS-P** algorithm to **automatically collect and annotate tens of thousands of step-wise actions** based on the five dimensions. This approach is applied across four distinct environment domains: Web, Android, Linux, and Windows. Unlike existing methods that rely on labor-intensive human annotations, this automated framework ensures scalability across diverse environments and generates a unified, fine-grained dataset that captures universal reasoning patterns, significantly reducing the cost and time required for data collection. **3)** Finally, using this dataset, we employ a **Triple-M** (multi-step, multi-objective, and multi-modal) **strategy to train a reward model**. This strategy integrates multiple dimensions of assessment and generates a synergistic gain by combining the strengths of five dimensions. As illustrated in Figure 1 (a), traditional methods focus solely on outcomes, require significant manual effort, and are coarse-grained, outcome-based, and labor-intensive. In contrast, our approach enables **Similar** to perform step-wise, multi-dimensional automatic assessment of agent trajectories, making it fine-grained, process-based, and autonomous. Building on this, **Similar** provides fine-grained rewards for GVA during the training phase, while seamlessly guiding GVA and robustly optimizing performance by scaling inference-time during the inference phase.

Since reward models are crucial for GVAs, and prior research has not focused on evaluating reward models, we propose **SRM**, the first benchmark in the GVA domain for step-wise, multi-dimensional reward model training and evaluation. Figure 1 (b) illustrates that it consists of 110k automatically annotated data points, divided into the scalable

*SRMTrain* (78k) for training **Similar** and the curated *SRMEval* (32k) for evaluating reward models.

Our reward model, **Similar**, can enhance the learning and reasoning of GVAs. **For training**, it serves as a reward model in a reinforcement learning framework, guiding GVAs to optimize its behavior based on action quality. By providing fine-grained feedback, it effectively guides the agents' learning process and enhances their performance. **For inference-time scaling**, it can be integrated with search algorithms such as Monte Carlo Tree Search (MCTS) to leverage reward signals for filtering candidate actions, and improve model performance (DeepSeek-AI et al., 2025; Zang et al., 2025). By selecting actions that are more likely to complete the task, it enhances accuracy and reduces time.

Extensive experiments demonstrate the superiority of our approach: **1)** Effectiveness of step-wise, multi-dimensional data: Using our collected data for reward modeling, **Similar**-RL-Llama achieves a 13.2% improvement over the baseline Llama-3.2-11B-Vision model on the *SRMEval* benchmark, demonstrating the effectiveness of our automated framework in enabling fine-grained assessment of GVA actions. **2)** Synergistic gain from the Triple-M strategy: The Triple-M strategy integrates multiple dimensions by leveraging the strengths of five dimensions, enabling **Similar**-TM-Llama to achieve an Avg score of 61.2 on *SRMEval*, significantly outperforming **Similar**-RL-Llama (53.9, a 13.5% improvement). This highlights the synergistic gain of our training strategy. **3)** Effective guidance in training and inference: **Similar** provides fine-grained, multi-dimensional feedback during training and integrates with search algorithms like MCTS to scale inference-time during inference to improve reasoning accuracy. Its strong performance across multiple benchmarks underscores its versatility and practical applicability.

Our contributions can be summarized as follows:

- We define five dimensions for step-wise GVA assess-

ment and an MCTS-P algorithm to collect fine-grained, cross-platform reward model data annotations.

- Based on these data, we propose a Triple-M strategy to train a reward model, called `Similar`, integrating multiple dimensions and generating synergistic gains for robust, fine-grained feedback.

- Moreover, we introduce *SRMEval*, a multi-step, multi-dimensional, and multi-platform benchmark for evaluating reward models, which is a set of *SRM* to advance research in reward model performance assessment.

- Experiments demonstrate that our approach, through step-wise multi-dimensional assessment, provides GVAs with superior intermediate signals during both training and inference-time scaling.

## 2. Related Work

### 2.1. Fine-Tuning Virtual Agent

Fine-tuning Virtual Agents traditionally relies on human-annotated datasets, which are labor-intensive and time-consuming (Wang et al., 2023). Methods such as imitation learning (Humphreys et al., 2022) and reinforcement learning (Branavan et al., 2009; 2010) have been employed to fine-tune agents based on curated expert trajectories or outcome rewards, but these approaches often suffer from compounding errors and limited exploration (Christianos et al., 2023; Xi et al., 2024; Song et al., 2024). Recent advancements, such as reject sampling fine-tuning (RFT) (Yuan et al., 2023) and direct policy optimization (DPO) (Rafailov et al., 2023), have sought to reduce reliance on human annotations by leveraging both successful and failure trajectories (Lai et al., 2024; Zhang et al., 2024c). However, these methods face significant challenges, including the lack of process supervision and reliance on human-annotated data, which limit their scalability and adaptability (Xu et al., 2021; He et al., 2024; Rawles et al., 2024b). In contrast, our work addresses these limitations by introducing a novel training paradigm that leverages multi-dimensional process supervision and automated annotation to enhance the learning and reasoning capabilities of GVAs.

### 2.2. Reward Models for Virtual Agent

Reward Models (RMs) are critical for guiding virtual agents by evaluating action quality (Zhai et al., 2024; Zhou et al., 2024a). While Outcome Reward Models (ORMs) focus on task success (Yu et al., 2024a; 2023a), Process Reward Models (PRMs) provide feedback on intermediate steps, offering an evaluation of agent performance in complex reasoning tasks (Uesato et al., 2022). Recent studies show that PRMs outperform ORMs in tasks like math reasoning, where process supervision is essential (Lightman et al., 2023; Li et al., 2023c). However, generating high-quality process supervision data remains challenging, as human

annotation is expensive. To address this, methods like step-level Q-value (Zhai et al., 2024) and ReST-MCTS* (Zhang et al., 2024b) have explored MCTS to automate data collection, achieving significant gains. Building on these insights, our work introduces a step-wise, multi-dimensional system leveraging MCTS to collect fine-grained annotations, enabling a robust reward model to guide GVAs.

## 3. Method

In this section, we present the pipeline for training our proposed `Similar` model. The *SRM* benchmark will be introduced in Section 4. As shown in Figure 2, to evaluate agent steps multi-dimensionally, we first define five-dimension process supervision (Section 3.1). Next, we introduce an MCTS-P algorithm to automatically collect step-wise annotations (Section 3.2). Finally, we design the Triple-M strategy to train `Similar`, achieving synergistic gains across five dimensions (Section 3.3).

### 3.1. Five-Dimensional Process Supervision Framework

To assess the quality of an agent's steps, we systematically define a five-dimensional process supervision framework. Given task complexity and interdependencies, a single metric is insufficient for assessing step quality (Zhai et al., 2024). Our framework addresses this limitation by covering the multi-faceted nature of step assessment. The first three dimensions—*Helpfulness*, *Odds of Success*, and *Efficiency*—are computed automatically, while the remaining two—*Task Relevance* and *Coherence*—are assessed using MLLMs. These dimensions are independent and interpretable, ensuring broad applicability across tasks.

The current step is denoted as $S_i$, where $i$ is the step index. The three automatic metrics are derived through MCTS simulations (Luo et al., 2024). For $S_i$, we simulate $N$ subsequent trajectories until a termination condition is met (i.e., the agent completes the task or reaches the maximum step length). We define the basic reward $r_i$ as:
$$r_i = \begin{cases} 1, & \exists a_{i,j} \in A,\ a_{i,j} = a^* \\ 0, & \text{otherwise} \end{cases}, \ j \in N, \text{ where } a^* \text{ repre-}$$
sents the ground truth, $a_{i,j}$ denotes the final action of the $j$-th trajectory in step $i$, and $A$ is the set of all actions. The following sections detail how each dimension assesses $S_i$.

*Helpfulness (H).* It quantifies whether a given step contributes positively or negatively to task completion, assigning values inversely proportional to the trajectory length. This dimension is designed to assess the impact of each step on the overall task trajectory. Steps that facilitate task completion are considered helpful, while those that hinder progress are assigned negative values. For example, each step in a 3-step successful trajectory is worth $1/3$, while steps hindering progress (those failing to lead to success) receive corresponding negative values. And in two successful trajectories of the same task, the steps in the trajectory with fewer steps will have higher *Helpfulness* value.

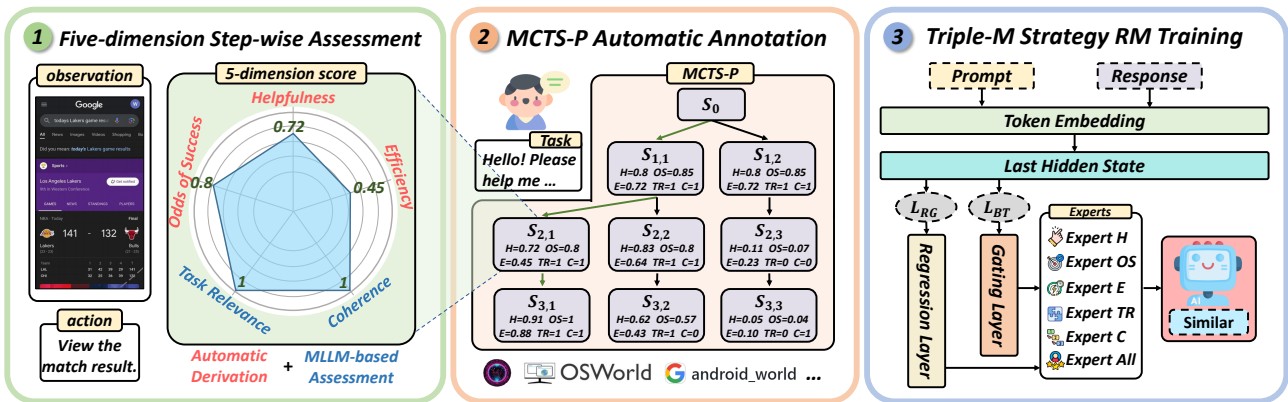

Figure 2: `Similar` **model training pipeline.** First, we systematically define five dimensions to describe the quality of an agent's step. Next, we propose an MCTS-P algorithm to automatically collect annotated step-wise data. Finally, we design the Triple-M strategy to train the `Similar` model, which can guide the agent during both the training and inference phases.

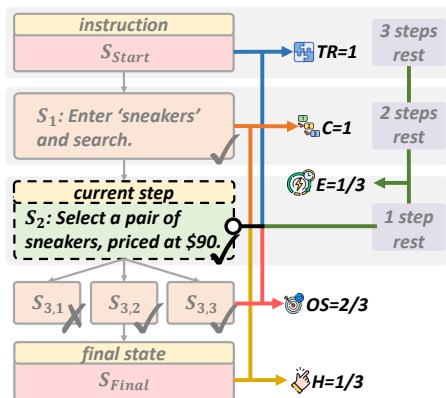

Figure 3: **An example describing the** 5 **dimensions**. A single step by an agent relates to 5 task elements: instruction, last step, next step, final state, and number of steps. Our 5 dimensions align with these: **H** (final state), **OS** (next step), **E** (number of steps), **TR** (instruction), and **C** (last step).

The *Helpfulness* can be calculated as the following formula:

$$H_i = \frac{1 - AC_{i-1}}{M - i + 1}(2r_i - 1),$$

where $AC_i = \begin{cases} 0, & i = 0 \\ \max(AC_{i-1} + H_i, 0), & \text{otherwise} \end{cases}$, which is a mathematical placeholder to recursively track cumulative *Helpfulness* scores during MCTS rollouts. And $M$ is the total number of reasoning steps.

***Odds of Success (OS).*** It measures the probability that a given step will lead to the successful completion of the task. This dimension is crucial for identifying steps that are more likely to result in a successful outcome. Steps with higher values are more likely to lead to success, while those with lower values are less likely to succeed. Conversely, incorrect steps lead to failure. The *Odds of Success* is calculated

by evaluating the proportion of successful paths among simulated results from a given step.

The formula for *Odds of Success* is defined as:

$$OS_i = \frac{\sum_{j=1}^{N} \mathbb{I}(a_{i,j} = a^*)}{N},$$

where $\mathbb{I}(\cdot)$ is the indicator function.

***Efficiency (E).*** It evaluates whether a given step is operationally efficient in terms of resource consumption, such as time or computational effort. A fundamental assumption is that fewer steps equate to higher efficiency, as shorter paths imply lower resource usage. Steps that reduce the total number of steps required to complete a task are considered efficient, as they enable the agent to accomplish the task more quickly and with fewer resources.

The *Efficiency* metric is calculated as the following formula:

$$E_i = \frac{Len_{i-1} - Len_i}{len_0},$$

where $Len_i = \text{avg}(Len(S_{i,j}))$, and $Len(S_{i,j})$ represents the number of steps remaining to complete the task after executing action $a_{i,j}$.

***Task Relevance (TR).*** It assesses whether a step is related to the task instruction. Some steps may be task-relevant but still fail (e.g., recording incorrect notes), while others may be irrelevant yet contribute to success (e.g., clicking on a blank screen). These distinctions cannot be captured through automated calculations. However, MLLMs with advanced image understanding can evaluate this dimension. *Task Relevance* is binary, with values $\{0, 1\}$.

***Coherence (C).*** It: measures the continuity and logical flow between consecutive steps. Some operations, although task-relevant, efficient, and likely to lead to success, may lack coherence with the previous step. For example, in a task

such as "Query the Lakers' game result and record it in a Note," opening a browser and a Note simultaneously may lack coherence compared to directly searching for the game result after opening the browser. *Coherence* is also evaluated using MLLMs and is a binary classification dimension, with possible values of $\{0, 1\}$.

The prompts of two MLLM-based assessment dimensions is detailed in Appendix A. To better understand the five dimensions, we use Figure 3. Each agent step relates to five task elements: instruction, last step, next step, final state, and step count. In the figure, step $S_2$ is assessed by these dimensions. The task requires 3 steps: $S_1$, $S_2$, and $S_{3,2}$, with all sharing an $H$ value of $\frac{1}{3}$. Among the next steps from $S_2$, $S_{3,2}$ and $S_{3,3}$ are correct, yielding an *OS* value of $\frac{2}{3}$. The $E$ value is $\frac{1}{3}$, calculated as $\frac{2-1}{3}$. Since $S_2$ aligns with $S_1$ and the instruction, *TR* and *C* values are 1.

## 3.2. Automatic Generalist Dataset Collecting

MCTS (Monte Carlo Tree Search) is a heuristic search algorithm used in decision-making, combining random sampling and tree-based search to find the optimal option. Based on its advantages, such as its scalability and efficient exploration-exploitation balance (Luo et al., 2024; Wang et al., 2024c), we propose a modified version, MCTS-P, to automatically collect annotated data. MCTS-P leverages the five dimensions introduced in Section 3.1 to comprehensively assess each step taken by the virtual agent.

In MCTS-P, the five-dimensional scores are used as the basis for node selection and backpropagation. Specifically, the algorithm computes a weighted sum of the five dimensions to obtain a composite score for each step. This composite score serves as the value $v_{i,j}$ for each node $S_{i,j}$ in the search tree. The tree structure in MCTS-P is similar to traditional MCTS, with each node $S_{i,j}$ storing the action $a_{i,j}$, visit count $n_{i,j}$, and value $v_{i,j}$. The detailed pseudo-code for MCTS-P is provided in Algorithm 1.

To build a comprehensive and generalist dataset for training and testing reward models, we collect a large number of task trajectories from agents across four different platforms: Web, Android, Linux, and Windows. Using the MCTS-P algorithm, we perform automatic data annotation to collect process supervision signals. The annotation process involves the following steps: **1)** For each node $S_{i,j}$ in the search tree $T_q$, we calculate the minimum number of steps $M$ required to reach a correct answer. **2)** During the expansion phase, the algorithm simulates $N$ possible outcomes for each step to obtain the basic reward $r_i$. **3)** Based on $M$ and $r_i$, we compute the three automatically calculated dimensions: *Helpfulness*, *Odds of Success*, and *Efficiency*. **4)** We then use a MLLM (GPT-4o (Hurst et al., 2024)) to evaluate the *Task Relevance* and *Coherence* of each step. **5)** We prune all incomplete branches (those that do not reach a final answer) and verify the correctness of the

---

**Algorithm 1** MCTS-P Algorithm

**Input:** Initial state $s_0$
1: Create root node $S_0$ with state $s_0$
2: **while** within computational budget **do**
3:    $S_i \leftarrow$ TreePolicy($S_0$)
4:    $\Delta \leftarrow$ DefaultPolicy($s(S_i)$)   // Simulate a random playout to estimate
5:    Backup($S_i, \Delta$)   // Backpropagate the value to update parent nodes
6: **end while**
7: **return** $a$(BestChild($S_0, 0$))
8:
9: **function** TreePolicy($S$)
10:   **while** $S$ is nonterminal **do**
11:     **if** $S$ not fully expanded **then**
12:      **return** Expand($S$)   // Expand the tree by adding a new child node
13:     **else**
14:      $S \leftarrow$ BestChild($S, C$)
15:     **end if**
16:   **end while**
17:   **return** $S$
18: **end function**
19:
20: **function** BestChild($S, c$)
21:   $v(S) = H(S) + OS(S) + E(S) + TR(S) + C(S)$
22:   **return** $\arg\max_{S' \in \text{children of } S} \left( \frac{v(S')}{n(S')} + c\sqrt{\frac{2\ln n(S)}{n(S')}} \right)$
23: **end function**
**Output:** Action $a$

---

remaining paths using the evaluation methods provided by the four benchmark environments. The obtained trajectories are selected as the final dataset for training and evaluation.

## 3.3. Triple-M Strategy for RM Training

Traditional reward modeling tasks typically rely on human-annotated data (Wang et al., 2024a), whereas our approach generates step-wise annotations across multiple dimensions. To address the challenge of integrating Multi-step, Multi-dimensional, and Multi-modal data, we propose a novel **Triple-M strategy** tailored for virtual agents.

Our Triple-M strategy leverages a pre-trained decoder-only MLLM as the backbone feature extractor $f_\theta$, divided into two stages. The first stage trains a regression layer for five-dimensional score prediction. For each input sequence $x \oplus y$ (where $x$ represents the prompt and $y$ represents the response), we extract the last hidden state $h \in \mathbb{R}^d$ with $d$-dimensional features and map it to a five-dimensional reward score through a linear regression layer $W \in \mathbb{R}^{d \times 5}$. The model is optimized using a regression loss:

$$L_{RG} = \min_{\theta, W} \mathbb{E}_{x,y,r \in \mathcal{D}} \|W^\top h - r\|_2^2,$$

where $r \in \mathbb{R}^5$ is the ground-truth reward vector, and $\mathcal{D}$ is the training dataset.

In the second stage, we train a gating network to dynamically balance the five-dimensional scores, addressing the multi-objective optimization problem. We introduce a prompt-aware gating network $g_\phi$, implemented as a shallow multi-layer perceptron (MLP). This network dynamically adjusts the model's focus based on the input prompt $x$. The gating network computes non-negative coefficients $w \in \mathbb{R}^5$ for the five reward dimensions. These coefficients are derived from the last hidden state corresponding to the prompt $x$ and normalized via a softmax function.

| Instruction | Observation | Step Idx | Trajectory | Type | Candidate Action Pair | |
|---|---|---|---|---|---|---|
| In Simple Calendar Pro, delete all the events. | | 2 | Step1: Click Search. | E | Click Calender. | Input text "Simple Calendar Pro". |

Figure 4: A data point of *SRMEval*.

The gating network is trained using the Bradley-Terry (BT) loss (Bradley & Terry, 1952) function, which aligns the model's predictions with human preferences. The BT loss is formulated as:

$$L_{BT} = \min_{\phi} \mathbb{E}\left[-\log \frac{\exp(R_{\text{chosen}})}{\exp(R_{\text{chosen}}) + \exp(R_{\text{rejected}})}\right],$$

where $R_{\text{chosen}}$ and $R_{\text{rejected}}$ represent the preference scores for the chosen and rejected responses. During training, only the gating network parameters are updated, while the backbone network and regression layer remain frozen.

Finally, the scalarized reward $R$ of the trained **Similar** model is computed as $R = g_{\phi}(f_{\theta}(x))^{\top}r$. Through this design, our **Similar** model can not only output five-dimensional scores but also a comprehensive score that balances the five dimensions, just like an all-around expert combining the capabilities of six experts.

## 4. *SRM* Benchmark

We introduce the *SRM* benchmark, built from multi-modal, multi-dimensional, and multi-platform annotated data.

**Data Collection.** We used GPT-4o-1120 (Hurst et al., 2024) as the agent to collect agent action trajectories across four benchmarks—WebArena (WA) (Zhou et al., 2024b), VisualWebArena (VWA) (Koh et al., 2024), Android World (AW) (Rawles et al., 2024a), and OSWorld (OW) (Xie et al., 2024). Since these environments do not provide dedicated training and test sets, to ensure fairness and prevent data leakage, we rigorously used $70\%$ data provided by these benchmarks for agent trajectory data collection, while the remaining $30\%$ were reserved for evaluation experiments to ensure no data overlap between *SRMTrain* and evaluation sets. Ultimately, we collected $10k$ agent trajectories by generating multiple distinct actions per step through task-specific prompt injection and stochastic exploration. And we constructed $110k$ preference pairs for the *SRM* benchmark based on the scores of the designed dimensions. We also sampled some data for human experts to verify the accuracy of the pairs, as detailed in Appendix C.

**Dataset Construction.** We carefully selected 32k data points for manual annotation as the test set *SRMEval*, while the remaining 78k data points were used as the training set *SRMTrain* to train the **Similar** model. The test data tasks are distinct from those in the training data. As shown in Figure 4, each data point in *SRMEval* includes instruction,

Table 1: Performance comparison of common MLLMs, **Similar**-RL, and **Similar**-TM on *SRMEval*.

| Reward Model | H | OS | E | TR | C | Tot | Traj | Avg |
|---|---|---|---|---|---|---|---|---|
| GPT-4-Turbo | 44.7 | 46.3 | 44.8 | 48.7 | 42.3 | 46.5 | 44.5 | 46.6 |
| GPT-4o | 49.9 | 50.1 | 47.9 | 51.4 | 43.8 | 51.1 | 49.8 | 51.4 |
| InternVL-2.5 | 38.9 | 43.1 | 44.3 | 41.4 | 41.8 | 40.7 | 39.0 | 40.9 |
| Qwen2-VL | 45.7 | 42.1 | 41.7 | 44.5 | 42.1 | 43.2 | 41.6 | 42.9 |
| + Similar-RL | 52.4 | 49.6 | 48.3 | 50.2 | 47.9 | 51.0 | 45.4 | 49.2 |
| + Similar-TM | 60.5 | 57.8 | 56.6 | 59.7 | 56.2 | 58.4 | 53.9 | 57.3 |
| Llama-3.2-V | 48.2 | 47.6 | 47.1 | 51.1 | 42.6 | 49.5 | 44.5 | 47.6 |
| + Similar-RL | 55.1 | 52.1 | 52.7 | 55.3 | 47.8 | 54.6 | 49.5 | 53.9 |
| + Similar-TM | 63.8 | 60.5 | 59.2 | 62.7 | 56.8 | 61.4 | 58.7 | 61.2 |

observation screenshot, step index, trajectory, evaluation type, and candidate action pair. The evaluation types include our proposed five key dimensions—*Helpfulness* (H), *Odds of Success* (OS), *Efficiency* (E), *Task Relevance* (TR), and *Coherence* (C)—as well as a total dimension that integrates the five dimensions (Tot, weighted sum of the five dimensions) and a trajectory-level dimension (Traj, average Tot score of all steps in trajectory). More visualization cases of *SRMEval* are detailed in Appendix F.

**New Task and Evaluation Metric.** Based on *SRMEval*, we proposed a new task for reward models in the virtual agent domain: *Selecting the better action from candidate action pair at step $i$ in a specific dimension*. The evaluation metric is Accuracy, measuring the reward model's ability to select the better action. Accuracy is calculated under each evaluation type. For clarity, we use abbreviations such as H to represent each metric in our experiments.

## 5. Experiments

### 5.1. Experimental Setup

**Baselines.** We selected two baseline methods: **1)** Qwen2-VL-7B-Instruct (Wang et al., 2024b) and Llama-3.2-11B-Vision-Instruct were directly used as reward models, with prompts provided (detailed in in Appendix A) to score agent steps. **2)** **Similar**-RL-Qwen and **Similar**-RL-Llama, whose backbones match the aforementioned models, were trained using reinforcement learning (Arulkumaran et al., 2017) on our *SRMTrain* dataset to score agent steps.

To benchmark against these baselines, we introduce **Similar**-TM-Qwen and **Similar**-TM-Llama, which are trained on the *SRMTrain* dataset using the Triple-M strategy with Qwen2-VL-7B-Instruct and Llama-3.2-11B-Vision-Instruct as backbones, respectively.

**Evaluation Benchmarks.** We first tested the preference alignment capability of **Similar** on our *SRMEval*, compared with GPT-4o-1120, GPT-4-Turbo, and InternVL-2.5-8B. Additionally, we evaluated our model's effectiveness as a reward model for virtual agents during both the training and inference phases. **1) Training Phase.** Using WebArena and Android World as benchmarks, we employed our model and other reward models to annotate GPT-4o-collected data

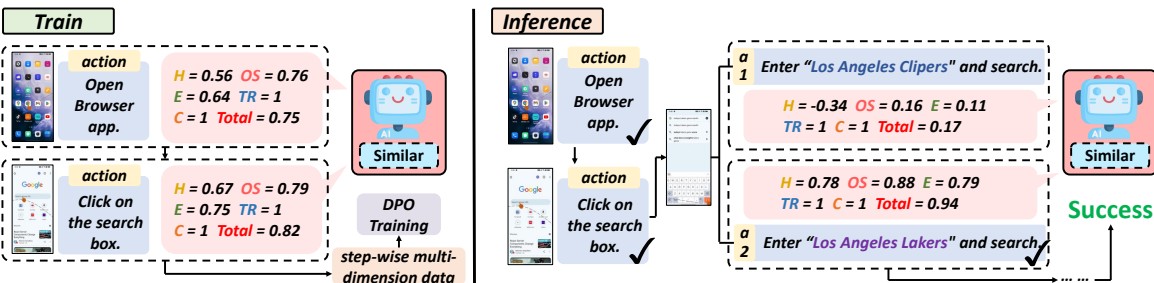

Figure 5: A case of **Similar** provides guidance for GVA training and inference.

Table 2: Task Success Rates (SR) on Android World and WebArena in training setting.

| AGENT | REWARD MODEL | ANDROID WORLD SR | WEBARENA SR |
|---|---|---|---|
| GPT-4-TURBO | / | 24.1 | 11.2 |
| GPT-4O | / | 25.4 | 12.7 |
| UGROUND | / | 32.4 | 19.6 |
| UGROUND | QWEN2-VL | 32.6 | 20.2 |
| UGROUND | + SIMILAR-RL | 33.1 | 26.5 |
| UGROUND | + SIMILAR-TM | 33.9 | 35.9 |
| UGROUND | LLAMA-3.2-V | 33.0 | 23.4 |
| UGROUND | + SIMILAR-RL | 33.8 | 29.6 |
| UGROUND | + SIMILAR-TM | **34.6** | **36.7** |
| OS-ATLAS | / | 30.4 | 20.2 |
| OS-ATLAS | QWEN2-VL | 30.8 | 20.8 |
| OS-ATLAS | + SIMILAR-RL | 32.1 | 25.9 |
| OS-ATLAS | + SIMILAR-TM | 34.2 | 34.5 |
| OS-ATLAS | LLAMA-3.2-V | 31.3 | 22.4 |
| OS-ATLAS | + SIMILAR-RL | 33.6 | 27.4 |
| OS-ATLAS | + SIMILAR-TM | **34.9** | **35.6** |

Table 3: Task Success Rates (SR) on Android World and OSWorld in inference setting.

| AGENT | REWARD MODEL | ANDROID WORLD SR | OSWORLD SR |
|---|---|---|---|
| GPT-4-TURBO | / | 24.1 | 8.4 |
| GPT-4-TURBO | QWEN2-VL | 24.9 | 8.9 |
| GPT-4-TURBO | + SIMILAR-RL | 25.9 | 10.5 |
| GPT-4-TURBO | + SIMILAR-TM | 28.3 | 13.4 |
| GPT-4-TURBO | LLAMA-3.2-V | 25.3 | 8.8 |
| GPT-4-TURBO | + SIMILAR-RL | 26.5 | 10.8 |
| GPT-4-TURBO | + SIMILAR-TM | **30.4** | **13.9** |
| GPT-4O | / | 25.4 | 10.8 |
| GPT-4O | QWEN2-VL | 26.0 | 11.3 |
| GPT-4O | + SIMILAR-RL | 27.1 | 12.9 |
| GPT-4O | + SIMILAR-TM | 32.9 | 14.3 |
| GPT-4O | LLAMA-3.2-V | 26.2 | 11.7 |
| GPT-4O | + SIMILAR-RL | 29.6 | 13.1 |
| GPT-4O | + SIMILAR-TM | **34.6** | **16.5** |
| OS-ATLAS | / | 30.4 | 14.3 |
| OS-ATLAS | QWEN2-VL | 30.9 | 14.8 |
| OS-ATLAS | + SIMILAR-RL | 32.0 | 15.4 |
| OS-ATLAS | + SIMILAR-TM | 34.5 | 16.4 |
| OS-ATLAS | LLAMA-3.2-V | 31.5 | 14.8 |
| OS-ATLAS | + SIMILAR-RL | 32.9 | 15.7 |
| OS-ATLAS | + SIMILAR-TM | **35.4** | **17.8** |

from these environments, generating preference data. This preference data was then used to perform DPO training on the open-source agents OS-Atlas (Wu et al., 2024c) and UGround (Gou et al., 2024), validating our model's ability to guide agents in the training phase. **2) Inference Phase.** With Android World and OSWorld as benchmarks, we used OS-Atlas as the open-source agent and GPT-4o-1120 and GPT-4-Turbo as the closed-source agents. During inference, **Similar** and other reward models evaluated the agent's simulated $N$ actions, providing rewards and updating the states of nodes in MCTS. Notably, $30\%$ of the examples partitioned from these benchmarks mentioned earlier were used as the evaluation data.

## 5.2. Effective Alignment of Preference

We first report the performance of the models on *SRMEval* in Table 1. The main findings are as follows: **1)** Effectiveness of step-wise, multi-dimensional, cross-platform data: Using our collected data for reward modeling, **Similar**-RL-Llama achieved an Avg score of 53.9, remarkably outperforming the baseline Llama-3.2-11B-Vision-Instruct with 47.6 ($\uparrow$ 13.2%) and surpassing closed-source models GPT-4o (51.4) and GPT-4-Turbo (46.6). It demonstrates that training reward models with our data enables fine-grained, step-based evaluation, providing a more compre-

hensive and accurate assessment of GVA action quality. **2)** Synergistic gain from the Triple-M strategy: **Similar**-TM-Llama achieved an Avg score of 61.2, significantly outperforming **Similar**-RL-Llama with 53.9 ($\uparrow$ 13.5%). And it achieved higher scores across all dimensions, with improvements such as H increasing from 48.2 to 63.8 ($\uparrow$ 32.3%) and E increasing from 47.1 to 59.2 ($\uparrow$ 25.6%). The **Similar**-TM-Qwen model showed similar performance. This highlights the effectiveness of our Triple-M strategy, leveraging the complementary expertise of each component to achieve synergistic gain. The experiments demonstrate our model's ability to align preferences.

## 5.3. **Similar** for RL Training

We used GPT-4o and multiple reward models to annotate reward data across benchmark environments. The annotated data was then used to train the final agent via DPO. The results, shown in Table 2, demonstrate that our model significantly improves agent learning: **1)** The **Similar**-RL model derived through Reward Modeling on the *SRMTrain* dataset outperforms the baseline. When using OS-Atlas as the agent, **Similar**-RL-Llama achieves improvements of

Table 4: Abaltion study (inference experiments). **Similar** in table represents **Similar**-TM-Llama.

| MODEL | DIMENSION | | | | | SUCCESS RATE | |
|---|---|---|---|---|---|---|---|
| | H | OS | E | TR | C | AW | WA |
| BACKBONE | | | | | | 30.4 | 20.6 |
| +H | ✓ | | | | | 32.5 | 26.1 |
| +OS | | ✓ | | | | 31.9 | 24.7 |
| +E | | | ✓ | | | 31.6 | 23.3 |
| +TR | | | | ✓ | | 31.1 | 21.6 |
| +C | | | | | ✓ | 30.9 | 21.0 |
| +OS,E | | ✓ | ✓ | | | 32.7 | 27.5 |
| +H,E | ✓ | | ✓ | | | 33.1 | 29.8 |
| +H,OS | ✓ | ✓ | | | | 33.4 | 31.4 |
| +TR,C | | | | ✓ | ✓ | 31.5 | 22.5 |
| +H,OS,E | ✓ | ✓ | ✓ | | | 34.3 | 35.9 |
| +OS,E,TR,C | | ✓ | ✓ | ✓ | ✓ | 33.1 | 33.9 |
| +H,E,TR,C | ✓ | | ✓ | ✓ | ✓ | 33.9 | 35.7 |
| +H,OS,TR,C | ✓ | ✓ | | ✓ | ✓ | 34.2 | 36.5 |
| +H,OS,E,C | ✓ | ✓ | ✓ | | ✓ | 34.7 | 37.2 |
| +H,OS,E,TR | ✓ | ✓ | ✓ | ✓ | | 35.1 | 37.7 |
| SIMILAR | ✓ | ✓ | ✓ | ✓ | ✓ | 35.4 | 38.2 |

$10.5\%$ ($30.4 \rightarrow 33.6$) and $7.3\%$ ($31.3 \rightarrow 33.6$) over the original OS-Atlas model and the setting using Llama-3.2V as the reward model, respectively, on Android World. On WebArena, the improvements are $35.6\%$ ($20.2 \rightarrow 27.4$) and $22.3\%$ ($22.4 \rightarrow 27.4$), respectively. **2)** The **Similar**-TM model performed best. With OS-Atlas, **Similar**-TM-Llama achieved improvements of $3.8\%$ ($33.6 \rightarrow 34.9$) and $29.9\%$($27.4 \rightarrow 35.6$) on Android World and WebArena, respectively, compared to **Similar**-RL-Llama. **3)** When using UGround as the agent or adopting Qwen2-VL as the baseline reward model, comparable performance can be observed. The consistent performance improvements across different models and environments demonstrate that our method enhances virtual agents' learning capabilities.

### 5.4. **Similar** for Inference-Time Scaling

During inference, we used various reward models to evaluate the agent's $N$ simulated actions, providing rewards and updating MCTS node states. Table 3 shows that our model effectively guides the agent: **1)** Consistent with the training setup, the **Similar**-RL model outperformed both the original agent without a reward model and the setting using MLLM as the reward model. With GPT-4o, **Similar**-RL-Llama achieved improvements of $16.5\%$ ($25.4 \rightarrow 29.6$) and $12.9\%$ ($26.2 \rightarrow 29.6$) on Android World for these two settings, respectively. A similar performance is observed on OSWorld. **2)** The **Similar**-TM model performed best. With GPT-4o, **Similar**-TM-Llama achieved improvements of $16.8\%$ ($29.6 \rightarrow 34.6$) and $25.9\%$ ($13.1 \rightarrow 16.5$) on Android World and OSWorld, respectively, compared to **Similar**-RL-Llama. **3)** When employing GPT-4-Turbo, GPT-4o, or OS-Atlas as the agent, or when using Qwen2-VL as the baseline reward model, we consistently observe similar model performance. It can be concluded that our

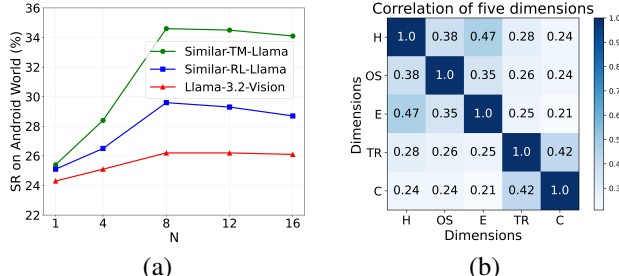

(a)    (b)

Figure 6: (a) Inference-time scaling research. The agent is GPT-4o. (b) Correlation research of five dimensions.

method is generalizable and effectively enhances the virtual agent's inference ability.

We further demonstrate that our model is essential for scaling the inference-time capabilities of agents by varying the number of child nodes $N$ in MCTS. As shown in Figure 6 (a): 1) When $N \leq 8$, agent performance improves. However, when $N > 8$, performance plateaus or declines, likely due to limitations in the agent model, as simulating more actions fails to identify viable paths. 2) **Similar**-RL and **Similar**-TM outperform other settings, with **Similar**-RL surpassing MLLM-based reward models and **Similar**-TM exceeding **Similar**-RL. These results demonstrate the superiority of our models while highlighting the challenges of scaling inference-time in agent systems.

### 5.5. Indepth Analysis

**Ablation study.** The results, shown in Table 4, show that models with partial-dimensional rewards underperformed compared to **Similar**. For example, on Android World, models excluding H, OS, E, TR, and C rewards showed declines of $6.9\%$, $4.4\%$, $3.5\%$, $2.0\%$, and $0.8\%$, respectively, with similar trends on WebArena. Analysis reveals that the H dimension has the most significant impact, as *Helpfulness* captures a step's contribution to task completion. The OS dimension follows closely, reflecting the influence of the current step on the next step. The C dimension has the least impact, as agent actions are often inherently coherent and contextually aligned. These results confirm that fine-grained rewards outperform coarse-grained ones and that our five dimensions comprehensively assess agent actions. More comprehensive results can be found in Appendix E.

**Case Study.** To demonstrate the role of our model in training and inference, we included visual cases, as shown in Figure 5. During training, **Similar** annotates the agent's trajectory with multi-dimensional scores, used for DPO training. During inference, the agent simulates multiple actions for a single step, and **Similar** evaluates these actions. In the figure, our model assigns high scores to action 2 at the third step, with a total score of $0.94$, while action 1 receives lower scores. Therefore, action 2, the highest-scoring action, is easily selected for the current step. More case studies are detailed in Appendix D.

**Correlation Study.** The Pearson correlation coefficients among the five dimensions are calculated to analyze their independence, as shown in Figure 6 (b). The results show that while some correlation exists among the five dimensions, the values are all below $0.47$, indicating independence.

## 6. Conclusion

In this work, we introduce a novel reward model-based paradigm for training GVAs. Our reward model, `Similar`, provides step-wise, multi-dimensional feedback during GVAs' training and inference, enabling fine-grained assessment. Additionally, we build the first reward model evaluation benchmark called *SRM*. Extensive experiments demonstrate our model's superior performance on *SRMEval* and its effectiveness in guiding GVAs across diverse tasks.

**Acknowledgment.** This work was supported by the NSFC (62272411), the Fundamental Research Funds for the Central Universities (226-2025-00017), the Key R&D Projects in Zhejiang Province (No. 2024C01106, 2025C01030), Ningbo Yongjiang Talent Introduction Programme(2024A-401-G),the Zhejiang NSF (LRG25F020001), Ant Group.

## Impact Statement

### Limitations and Future Investigation

While `Similar` demonstrates significant advancements in the training and inference of GVAs, several limitations remain. First, the current framework relies heavily on the quality of the automatically annotated data generated by the MCTS-P algorithm. Although this approach reduces human annotation efforts, it may introduce biases or inaccuracies in the reward signals, particularly in complex or ambiguous task scenarios. Future work should focus on improving the robustness of the automatic annotation process, potentially by incorporating more sophisticated error-correction mechanisms or hybrid human-AI annotation strategies.

Moreover, the scalability of `Similar` across diverse environments and tasks is promising but not yet fully explored. While the current experiments cover four major platforms (Web, Android, Linux, and Windows), the model's performance in more niche or specialized domains remains untested. Future investigations should extend the evaluation to a broader range of environments, including those with less structured or more dynamic interfaces, to ensure the generalizability of the approach.

### Impact on RL Training

`Similar` has a transformative impact on reinforcement learning (RL) training for GVAs. By providing fine-grained, multi-dimensional feedback, `Similar` enables more efficient and effective learning compared to traditional outcome-based reward models. The step-wise, multi-dimensional assessment allows the agent to identify and correct errors at intermediate stages, leading to faster convergence and improved task performance.

The Triple-M strategy further enhances the RL training process by integrating multiple dimensions of assessment and leveraging the strengths of different experts. This synergistic approach not only improves the accuracy of the reward signals but also ensures that the agent learns robust policies that generalize well across diverse tasks and environments. As a result, `Similar` significantly reduces the reliance on labor-intensive human annotations, making RL training more scalable and cost-effective.

### Impact on Inference-Time Scaling

Recent outstanding work has demonstrated that inference-time scaling can significantly enhance agent performance (DeepSeek-AI et al., 2025; Wu et al., 2024b). `Similar` significantly enhances the inference-time capabilities of GVAs by integrating with search algorithms like MCTS. During inference, our model provides fine-grained, multi-dimensional rewards to evaluate and filter candidate actions, ensuring the agent selects the most promising paths. This not only improves task completion accuracy but also reduces computational overhead, making it highly efficient for real-time applications.

Experiments demonstrate that our model consistently outperforms baseline models and MLLM-based reward settings across diverse environments, such as Android World and OSWorld. The `Similar`-TM model, leveraging the Triple-M strategy, achieves the best performance, highlighting the synergistic gains from integrating multiple dimensions of assessment. Furthermore, our model effectively scales inference-time computations, with performance improvements observed when expanding the number of child nodes in MCTS, though performance plateaus beyond a certain threshold due to inherent agent limitations.

### Impact on Data Cleaning

`Similar` also has significant implications for data cleaning in the context of GVA training. The model's ability to provide fine-grained, multi-dimensional feedback allows for the precise identification and systematic removal of low-quality or noisy data points. This is particularly useful in large-scale datasets where manual inspection is impractical. By filtering out irrelevant or incoherent actions and prioritizing consistent, task-aligned examples, our model ensures that the training data is of high quality, leading to more robust and reliable agent performance.

Moreover, the automatic annotation process introduced by `Similar` reduces the need for human intervention in data cleaning, further enhancing the scalability of GVA training. This is especially beneficial in domains where data is abundant but of varying quality, such as web navigation or mobile app interaction. By improving the quality of the training data, our model contributes to the overall efficiency and effectiveness of GVA training pipelines.

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

# Appendix

This is the Appendix for the paper "Boosting Virtual Agent Learning and Reasoning: A Step-wise, Multi-dimensional, and Generalist Reward Model with Benchmark".

## Overview

In this supplementary material we present:

- Detailed prompts for testing general MLLMs on *SRMEval* and for building *SRM* are provided in Section A.

- Pseudocode for the **Similar** pipeline, including key algorithmic steps, is described in Section B.

- Human evaluation details and human acceptance of 5-dimension data from *SRM* are illustrated in Section C.

- Case studies demonstrating the applications of **Similar** are presented in Section D.

- Comprehensive ablation experiments are conducted and analyzed in Section E.

- Additional visualizations of *SRMEval*, showcasing task performance, are provided in Section F.

## A. Detailed Prompt Design

This section provides a comprehensive overview of the prompt designs used in our experiments. We detail the prompts for testing general MLLMs on *SRMEval* and the prompts for building the *SRM* model, highlighting their structure and purpose.

### A.1. Prompt for testing general MLLMs on *SRMEval*

This subsection describes the prompts used to evaluate general MLLMs on the *SRMEval* benchmark.

---

**Prompt for *SRMEval* (main part)**

```
You are an expert in evaluating the
performance of a Virtual Agent.

The Virtual Agent is designed to help
a human user complete specified tasks
(such as app usage, web navigation, web
content Q&A, etc.) on various platform
applications (such as websites, mobile
devices, operation systems, etc.) based
on given instructions. Given the user's
INSTRUCTION, the OBSERVATION of current
platforms, the action TRAJECTORY of the
```

---

```
agent, the two ACTION_X and ACTION_Y
predicted by the agent, and the current
action step number STEP_IDX. Your GOAL is
to help me complete step-wise evaluation,
that is, evaluate the quality of the
Agent's ACTION in a specific dimension.
Choose the better action (ACTION_X or
ACTION_Y) based on the given ⟨EVALUATION
DIMENSION⟩. Output ''Y'' and the reason
if ACTION_X is better, or ''X'' and the
reason if ACTION_Y is better. Do not
output responses like ''two actions are
similar''.
```

⟨**Word Meaning**⟩
```
1.INSTRUCTION: refers to the command of
human users to the Agent, which is the
specific content that the Agent needs to
complete the task on a specific platform,
that is, the ultimate GOAL of the Agent.
2.OBSERVATION: refers to the specific
information of the current platform that
an agent can observe on the platform
where the task needs to be completed,
which is the environment in which the
agent is currently located. In our task,
observations are presented in the form
of images, known as screenshots.
3.TRAJECTORY: refers to the action
prediction made by an agent in the
past to complete the INSTRUCTION, which
records all actions taken by the agent
from the first step to the current step.
If this is the first step, then the
trajectory is empty.
4.ACTION: refers to the predicted
operation of the Agent in the current
state to complete the INSTRUCTION in the
current step. This operation generally
refers to a simple action command, such
as ''CLICK'', ''TYPE'', etc. Note
that ACTION is the result predicted by
the agent after observing the current
OBSERVATION, and the Agent often cannot
complete the task in one step.
5.STEP_IDX: refers to the sequence number
of the Agent executing the current
ACTION to complete the INSTRUCTION.
```

Here is the evaluation dimension part of the prompts.

*Helpfulness*.

> **Prompt for *SRMEval* (Helpfulness)**
>
> 1.**[HELPFULNESS]**
> 1.1 Meaning: It indicates the degree
> to which this step contributes to the
> completion of the final task. There are
> good and bad contributions, the correct
> steps will give a positive contribution,
> and the wrong steps will give a negative
> contribution.
> 1.2 Design motivation: Different steps
> contribute differently to the completion
> of the final task, with good steps
> helping to accomplish the task and bad
> steps hindering it. Good steps should
> be rewarded positively, while bad steps
> should be punished negatively. If each
> step is correct and the total number
> of steps is 5, then the contribution
> of each step can be considered as 1/5,
> meaning that each step completes 1/5
> of the final task. If 4 more steps
> are needed from the current step and
> the current step is incorrect, then the
> contribution of the current step is -1/4,
> indicating that it hinders 1/4 of the
> final task progress.

*Odds of Success*.

> **Prompt for *SRMEval* (Odds of Success)**
>
> 2.**[ODDS OF SUCCESS]**
> 2.1 Meaning: It indicates the potential
> of the step to complete the task, which
> is the probability of a step reaching
> the completion of the task. 2.2 Design
> motivation: The more correct steps lead
> to a higher probability of success in
> the final task, and the more incorrect
> steps lead to a higher probability of
> failure in the final task. Different
> steps have different potential to
> complete the task. If one step of the
> agent is to follow the Instructions
> to complete the task, then this step
> generally has high potential. We can
> derive the probability of a step leading
> to success from the N paths generated by
> that step, which serves as the potential
> for that step to complete the task which
> is crucial for evaluating.

*Efficiency*.

> **Prompt for *SRMEval* (Efficiency)**
>
> 3.**[EFFICIENCY]**
> 3.1 Meaning: It indicates whether this
> step is efficient in completing the
> task. We calculate this metric as the
> difference between 'the number of steps
> required to complete the final task
> after the current step' and 'the number
> of steps required to complete the final
> task after the previous step', divided
> by 'the total number of steps required
> to complete the task'. This indicates
> the degree of efficiency improvement in
> completing tasks after the current step
> is executed.
> 3.2 Design motivation: A basic
> assumption is that the fewer steps the
> Agent operates, the more efficient it
> is, because the consumption of these
> paths (time consumption, hardware
> consumption) can be considered to be the
> least and the efficiency is the highest.
> Therefore, if the operation of a step
> can reduce the number of steps required
> to complete the task as a whole, then
> it can be considered that the operation
> of this step is very efficient. For
> example, after the previous step, it
> takes 7 steps to complete the task, but
> after the current step, it only takes
> 4 steps to complete the task. The
> difference of 7-4=3 is the efficiency
> improvement of the current step in
> completing the final task.

*Task Relevance*.

> **Prompt for *SRMEval* (Task Relevance)**
>
> 4.**[TASK RELEVANCE]**
> 4.1 Meaning: It indicates is whether
> the operation of the Agent is related to
> achieving the **INSTRUCTION**.
> 4.2 Design motivation: Some operational
> steps may prevent the task from being
> completed, but they are related to the
> task (for example, we need to ask the
> agent to take notes, and the agent takes
> notes, which is related to the task, but
> the recorded note content is incorrect,
> indicating that this is an incorrect
> step). Some operational steps may be

```
meaningless, but they can still lead
to task completion (such as clicking
on a blank screen without generating
any response, which is unrelated to the
task, but the agent's subsequent actions
can still result in task success).
Therefore, an indicator is needed to
identify whether the current step of
operation is related to the task.
4.3 Range of values after mapping: {0,
1}.  The larger the value, the greater
the correlation between the step and the
task.
```

*Coherence.*

**Prompt for *SRMEval* (Coherence)**

```
5.[COHERENCE]
5.1 Meaning:  It represents the
compactness and coherence between the
current step and the previous step.
5.2 Design motivation:  Some operations,
although task-related, not inefficient,
and highly likely to lead to success,
lack coherence with the previous step.
For example, the task is to ''query the
Lakers' game results and record them in
the Note''.  The Agent operations are as
follows:  a Open the browser; b.  Open
Note; c.  Create new notes; d.  Search
for Lakers games; e.  Query the results
of the competition; f.  Record the
results of the competition in your notes.
It can be found that the operations
of a and b lack coherence, and it is
more in line with human preferences to
directly search for competition results
after opening the browser instead of
simultaneously opening Note.
5.3 Range of values after mapping:  {0,
1}.  The larger the value, the greater
the coherence of the step.
```

Total dimension and Trajectory-level dimension.

**Prompt for *SRMEval* (Total and Trajectory-level)**

```
6.[TOTAL]
Meaning:  Integrated decision-making
based on the 5 dimensions mentioned
earlier.

7.[TRAJECTORY]
Meaning:  Represents the quality of
the entire trajectory, which can be
```

```
expressed as the average total score
of all steps in the trajectory.
```

### A.2. Prompt for building *SRM*

This subsection outlines the prompts designed for constructing the *SRM* model.

**Prompt for building *SRM***

```
You are a virtual agent.

The Virtual Agent is designed to help
a human user complete specified tasks
(such as app usage, web navigation, web
content Q&A, etc.)  on various platform
applications (such as websites, mobile
devices, operation systems, etc.)  based
on given instructions.

You will predict the next action based
on the following content [INSTRUCTION],
[OBSERVATION], [REASON_STEPS]:

1.[INSTRUCTION]: It is your ultimate
GOAL, and all your actions are aimed at
completing this task.
2.[OBSERVATION]: It is an observation of
an image, which is the screenshot of the
platform (such as a computer screen).
3.[REASON_STEPS]: They are the trajectory
of the actions you performed in the past
to complete the instruction, from which
you can understand how you thought in
order to complete the instruction.  If
it is empty, it means it is currently
the first step.
```

## B. `Similar` Pipeline Pseudocode

The rapid advancement of MLLMs (Li et al., 2020; 2022; Yu et al., 2023b; 2024b; Pan et al., 2024a; Wu et al., 2024a; Fei et al., 2024; Miao et al., 2024), which excel at integrating text, vision, and other modalities, has enabled the development of GVAs (Gao et al., 2024; Zhang et al., 2024a; Shen et al., 2023), which also inspires us to study Reward Models for GVAs with its Benchmark.

In this section, we present the pipeline pseudocode for training our proposed `Similar` model. The pipeline consists of three main components: 1) a five-dimensional process supervision framework to evaluate agent steps, 2) an automatic generalist dataset collecting process, and 3) a Triple-M strategy for reward model training.

### B.1. Five-Dimensional Process Supervision Framework

The five-dimensional process supervision framework systematically evaluates the quality of an agent's steps using

**Algorithm 2** Five-Dimensional Process Supervision

1: **Input:** Step $S_i$, ground truth $a^*$, number of simulations $N$
2: **Output:** Five-dimensional scores $(H_i, OS_i, E_i, TR_i, C_i)$
3: **Compute Helpfulness (H):**
4: $H_i = \frac{1-AC_{i-1}}{M-i+1}(1-2r_i)$
5: $AC_i = \max(AC_{i-1} + H_i, 0)$
6: **Compute Odds of Success (OS):**
7: $OS_i = \frac{\sum_{j=1}^{N} \mathbb{I}(a_{i,j}=a^*)}{N}$
8: **Compute Efficiency (E):**
9: $E_i = \frac{Len_{i-1}-Len_i}{len_0}$
10: $Len_i = \text{avg}(Len(S_{i,j}))$
11: **Compute Task Relevance (TR) and Coherence (C):**
12: $TR_i = \text{MLLM\_Evaluate}(S_i, \text{instruction})$
13: $C_i = \text{MLLM\_Evaluate}(S_i, S_{i-1})$
14: **return** $(H_i, OS_i, E_i, TR_i, C_i)$

---

**Algorithm 3** Automatic Generalist Dataset Collecting

1: **Input:** Task instruction $q$, platforms $\mathcal{P} = \{$Web, Android, Linux, Windows$\}$
2: **Output:** Annotated dataset $\mathcal{D}$
3: Initialize empty dataset $\mathcal{D}$
4: **for** each platform $p \in \mathcal{P}$ **do**
5:     Initialize MCTS-P tree $T_q$ for task $q$ on platform $p$
6:     **for** each node $S_{i,j}$ in $T_q$ **do**
7:         Calculate minimum steps $M$ to reach a correct answer
8:         Simulate $N$ trajectories to compute basic reward $r_i$
9:         Compute $H_i, OS_i, E_i$ using formulas from Algorithm 2
10:         Evaluate $TR_i$ and $C_i$ using MLLM (e.g., GPT-4)
11:         **if** node $S_{i,j}$ leads to a complete trajectory **then**
12:            Verify correctness using platform-specific evaluation methods
13:            Add annotated step $(S_{i,j}, H_i, OS_i, E_i, TR_i, C_i)$ to $\mathcal{D}$
14:         **end if**
15:     **end for**
16: **end for**
17: Prune incomplete branches from $T_q$
18: **return** Annotated dataset $\mathcal{D}$

---

five distinct dimensions: *Helpfulness (H)*, *Odds of Success (OS)*, *Efficiency (E)*, *Task Relevance (TR)*, and *Coherence (C)*. The following pseudocode outlines the computation of these dimensions for a given step $S_i$, providing a comprehensive assessment of step quality.

### B.2. Automatic Generalist Dataset Collecting

The automatic generalist dataset collecting process leverages the MCTS-P algorithm to collect annotated step-wise data across multiple platforms, including Web, Android, Linux, and Windows.

### B.3. Triple-M Strategy for Reward Model Training

The Triple-M strategy integrates multi-step, multi-dimensional, and multi-modal data for reward model training, ensuring high flexibility, robustness, and adaptability. The following pseudocode outlines the two-stage training process, which includes regression layer optimization and dynamic gating network adjustment.

## C. Human Evaluation Details and Human Acceptance of *SRM*

To ensure high-quality annotations, we collaborated with a professional commercial data labeling team. The process included: **1) Training Phase:** Annotators underwent three

---

**Algorithm 4** Triple-M Strategy for Reward Model Training

1: **Input:** Training dataset $\mathcal{D}$, pre-trained MLLM $f_\theta$, gating network $g_\phi$
2: **Output:** Trained reward model $R$
3: **Stage 1: Regression Layer Training**
4: **for** each batch $(x, y, r) \in \mathcal{D}$ **do**
5:     Extract hidden state $h = f_\theta(x \oplus y)$
6:     Compute predicted scores $\hat{r} = W^\top h$
7:     Update $\theta, W$ using $L_{RG} = \|\hat{r} - r\|_2^2$
8: **end for**
9: **Stage 2: Gating Network Training**
10: **for** each batch $(x, y_{\text{chosen}}, y_{\text{rejected}}) \in \mathcal{D}$ **do**
11:     Compute coefficients $w = g_\phi(f_\theta(x))$
12:     Compute preference scores $R_{\text{chosen}} = w^\top r_{\text{chosen}}$
13:     Compute preference scores $R_{\text{rejected}} = w^\top r_{\text{rejected}}$
14:     Update $\phi$ using $L_{BT} = -\log \frac{\exp(R_{\text{chosen}})}{\exp(R_{\text{chosen}})+\exp(R_{\text{rejected}})}$
15: **end for**
16: **return** Trained reward model $R = g_\phi(f_\theta(x))^\top r$

---

rounds of iterative "label-review-feedback" cycles to clarify ambiguities of annotation (e.g., the complexity of UI interaction tasks). Only after achieving $> 95\%$ accuracy on validation samples did formal annotation begin. **2) Formal Annotation:** Each test sample in *SRMEval* was independently labeled by three annotators and three checkers. The final data in test set required $> 99\%$ accuracy.

To validate the quality of our five-dimensional assessment data and ensure alignment with human preferences, we randomly sampled a batch of data from the *SRM* Benchmark. The sample size varied across dimensions due to differences in score distributions. This stems from their fundamental design - for instance, *Task Relevance* and *Coherence* are binary values, which naturally yield fewer possible preference pairs. Human annotators were then asked to select the better action from candidate action pairs in the sampled data, based on specific evaluation types. If the annotator considers a sample correct, mark it as 1; otherwise, mark it as 0, and calculate Accuracy as Human Acceptance.

The results, as shown in Table 5, demonstrate that the human acceptance rate for all five dimensions exceeds $78.8\%$, strongly indicating the superiority of our designed annotation dimensions and the high quality of the collected data.

The evaluation process was further enhanced by incorporating a rigorous double-blind annotation protocol, where neither the annotators nor the analysts were aware of the origin or automated scores of the candidate actions.

Table 5: Sample size and human acceptance rate for each dimension in *SRM*.

| Dimension | Sample Size | Human Acceptance |
|---|---|---|
| Helpfulness | 6000 | 87.9% |
| Odds of Success | 2000 | 78.8% |
| Efficiency | 6000 | 82.6% |
| Task Relevance | 1000 | 84.7% |
| Coherence | 2000 | 93.5% |

# D. Case Studies on `Similar` Applications

## D.1. *Helpfulness*

---

**Case of *Helpfulness***

► **Input:**

```
[INST]
Create a timer with 0 hours, 16
minutes, and 35 seconds. Do not start
the timer.
[/INST]

[OBS]
```

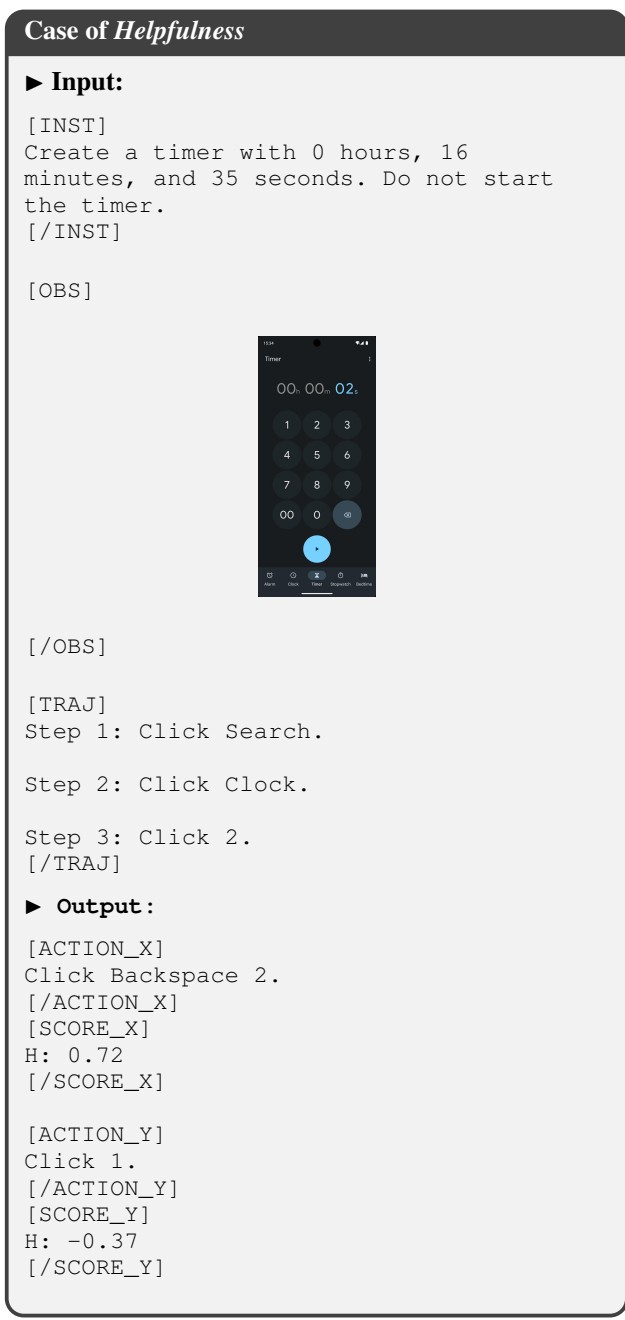

```
[/OBS]

[TRAJ]
Step 1: Click Search.

Step 2: Click Clock.

Step 3: Click 2.
[/TRAJ]
```

► **Output:**

```
[ACTION_X]
Click Backspace 2.
[/ACTION_X]
[SCORE_X]
H: 0.72
[/SCORE_X]

[ACTION_Y]
Click 1.
[/ACTION_Y]
[SCORE_Y]
H: -0.37
[/SCORE_Y]
```

---

In this case, the task is to set a timer for 16 minutes and 35 seconds. According to the reasoning steps, the previous action was clicking "2", which does not match the required time. The current step should involve deleting the incorrect input immediately. ACTION_X (Backspace) is correct and more helpful, while ACTION_Y (Click 1) further hinders task completion and is clearly harmful. Therefore, ACTION_X receives a higher *Helpfulness* score.

## D.2. *Odds of Success*

---

**Case of *Odds of Success***

► **Input:**

```
[INST]
In Simple Calendar Pro, delete all the
calendar events on 2023-10-27.
[/INST]

[OBS]
```

```
[/OBS]

[TRAJ]
Step 1: Click Search.

Step 2: Click Calendar.

Step 3: Scroll down.

Step 4: Long press 27 Friday.

Step 5: Click More options.
[/TRAJ]
```

► **Output:**

```
[ACTION_X]
Navigate back.
[/ACTION_X]
[SCORE_X]
OS: 0.75
[/SCORE_X]

[ACTION_Y]
Click Import events from a .ics file.
[/ACTION_Y]
[SCORE_Y]
OS: 0.13
[/SCORE_Y]
```

---

In this case, the task is to delete events on a specific day in the Calendar. The previous step, clicking "More Options", aimed to locate the delete button, but the current observation shows no delete option is available. Therefore, the correct action is to navigate back and search for the delete button elsewhere. ACTION_X (Navigate back) is the appropriate choice, while ACTION_Y (Import a file) is clearly incorrect. Thus, ACTION_X receives a higher *Odds of Success* score.

**D.3.** *Efficiency*

---
**Case of *Efficiency***

▶ **Input:**

[INST]
In Simple Calendar Pro, delete all the
events.
[/INST]

[OBS]

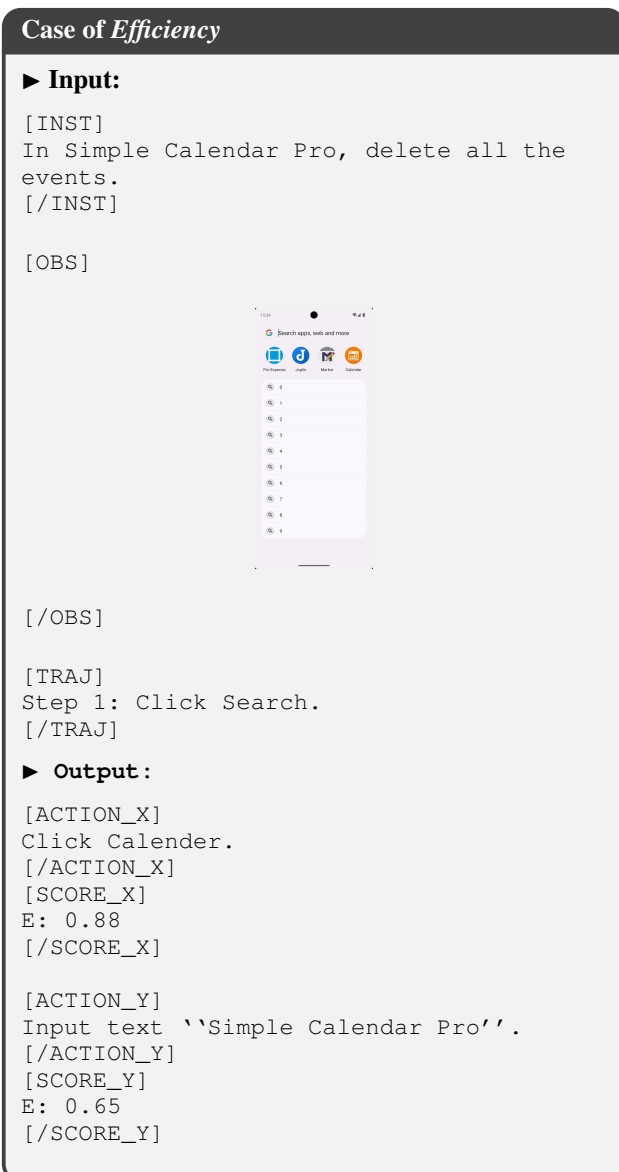

[/OBS]

[TRAJ]
Step 1: Click Search.
[/TRAJ]

▶ **Output:**

[ACTION_X]
Click Calender.
[/ACTION_X]
[SCORE_X]
E: 0.88
[/SCORE_X]

[ACTION_Y]
Input text ``Simple Calendar Pro''.
[/ACTION_Y]
[SCORE_Y]
E: 0.65
[/SCORE_Y]

---

In this case, since the search interface displays past search history, including "Calendar", ACTION_X (Click "Calendar") is more efficient in the *Efficiency* dimension. However, ACTION_Y is also a correct approach, so its *Efficiency* score remains relatively high.

**D.4.** *Task Relevance*

---
**Case of *Task Relevance***

▶ **Input:**

[INST]
Add these recipes to the Broccoli app:

1. Chicken Alfredo Pasta
    – Description: A healthy, delicious

---

meal.
    – Servings: 2
    – Prep Time: 10 mins
    – Ingredients: As desired
    – Directions: Cook pasta, toss with
Alfredo sauce and grilled chicken. Top
with Parmesan and spices.

2. Quinoa Salad with Vegetables
    – Description: Quick and easy for
busy days.
    – Servings: 8
    – Prep Time: 30 mins
    – Ingredients: To your liking
    – Directions: Mix quinoa, diced
veggies, feta, and lemon olive oil
dressing. Add spices for flavor.

3. Butternut Squash Soup
    – Description: A healthy, delicious
choice.
    – Servings: 1
    – Prep Time: 45 mins
    – Ingredients: Per taste
    – Directions: Saute onions and
garlic, add squash and broth. Puree and
season with nutmeg, salt, and pepper.
Substitute as needed.
[/INST]

[OBS]

[/OBS]

[TRAJ]
Step 1: Click Search.

Step 2: Input text Broccoli.

Step 3: Click Broccoli.

Step 4: Click New Recipe.

Step 5: Input text Chicken Alfredo
Pasta.

Step 6: Input text A delicious and
healthy choice for any time of the day.

Step 7: Input text 2 servings.

Step 8: Input text 10 mins.

```
Step 9: Input text as desired.

Step 10: Scroll down.

Step 11: Input text Cook fettuccine
pasta, toss with Alfredo sauce and
grilled chicken strips. Serve with a
sprinkle of Parmesan cheese. Try adding
a pinch of your favorite spices for
extra flavor.

Step 12: Click SAVE.
[/TRAJ]

▶ Output:

[ACTION_X]
Click Cook.
[/ACTION_X]
[SCORE_X]
TR: 1
[/SCORE_X]

[ACTION_Y]
Navigate back.
[/ACTION_Y]
[SCORE_Y]
TR: 0
[/SCORE_Y]
```

In this case, the task is to create a recipe. The first 12 steps have completed the recipe creation process. In the Broccoli app, the next action should be to directly click "Cook" (i.e., ACTION_X), which is highly relevant to the instruction. In contrast, "Navigate back" (i.e., ACTION_Y) is not directly related to the instruction. Therefore, ACTION_X receives a higher *Task Relevance* score.

### D.5. *Coherence*

**Case of *Coherence***

▶ **Input:**

```
[INST]
Add this exact product to my shopping
cart. I think it is in the "Herbs,
Spices \& Seasonings" category.
[/INST]

[OBS]
```

```
[/OBS]

[TRAJ]
Step 1: Click menuitem '\ue622 Grocery
& Gourmet Food' hasPopup: menu.

Step 2: Click link 'Pantry Staples(
4891 item )'.
[/TRAJ]

▶ Output:

[ACTION_X]
Click link 'Herbs, Spices \&
Seasonings( 707 item )'.
[/ACTION_X]
[SCORE_X]
C: 1
[/SCORE_X]

[ACTION_Y]
Click menuitem '\ue622 Grocery &
Gourmet Food' hasPopup: menu.
[/ACTION_Y]
[SCORE_Y]
C: 0
[/SCORE_Y]
```

In this case, the task is to find relevant products in the "Herbs, Spices & Seasonings" category. The previous step involved the agent entering a link, and the current step should logically advance the task. Clearly, ACTION_X (clicking the "Herbs, Spices & Seasonings" link) is a coherent and logical continuation, while ACTION_Y (repeating the action from step 1) is unintelligible and incoherent. Therefore, ACTION_X receives a higher *Coherence* score.

## E. Comprehensive Ablation Experiments

The extended ablation study, presented in Table 6, provides a more comprehensive analysis of the impact of each dimension in the **Similar** model. The results confirm the trends observed in the main experiments and offer additional insights into the contributions of the five dimensions—*Helpfulness (H)*, *Odds of Success (OS)*, *Efficiency (E)*, *Task Relevance (TR)*, and *Coherence (C)*—across three benchmarks: Android World, WebArena, and OSWorld.

### E.1. Impact of Individual Dimensions

The results demonstrate that the *Helpfulness (H)* dimension has the most significant impact on performance, consistent with the findings in the main experiments. For example, adding *H* alone improves the success rate on Android World from 30.7% to 32.5%, on WebArena from 20.6% to 26.1%, and on OSWorld from 14.6% to 15.8%. This aligns with our hypothesis that the quality of a step is primarily reflected in its contribution to task completion, which *H* effectively

Table 6: Ablation study (inference experiments). `Similar` in table represents `Similar`-TM-Llama.

| MODEL | DIMENSION | | | | | SUCCESS RATE | | |
| --- | --- | --- | --- | --- | --- | --- | --- | --- |
| | H | OS | E | TR | C | ANDROID WORLD | WEBARENA | OSWORLD |
| BACKBONE | | | | | | 30.4 | 20.6 | 14.3 |
| +H | ✓ | | | | | 32.5 | 26.1 | 15.8 |
| +OS | | ✓ | | | | 31.9 | 24.7 | 15.4 |
| +E | | | ✓ | | | 31.6 | 23.3 | 15.2 |
| +TR | | | | ✓ | | 31.1 | 21.6 | 14.9 |
| +C | | | | | ✓ | 30.9 | 21.0 | 14.8 |
| +H,OS | ✓ | ✓ | | | | 33.4 | 31.4 | 16.7 |
| +H,E | ✓ | | ✓ | | | 33.1 | 29.8 | 16.5 |
| +H,TR | ✓ | | | ✓ | | 32.8 | 28.5 | 16.3 |
| +H,C | ✓ | | | | ✓ | 32.6 | 28.2 | 16.2 |
| +OS,E | | ✓ | ✓ | | | 32.7 | 27.5 | 16.3 |
| +OS,TR | | ✓ | | ✓ | | 32.3 | 26.8 | 16.0 |
| +OS,C | | ✓ | | | ✓ | 32.1 | 26.5 | 15.9 |
| +E,TR | | | ✓ | ✓ | | 31.8 | 25.9 | 15.7 |
| +E,C | | | ✓ | | ✓ | 31.7 | 25.7 | 15.6 |
| +TR,C | | | | ✓ | ✓ | 31.5 | 22.5 | 15.1 |
| +H,OS,E | ✓ | ✓ | ✓ | | | 34.3 | 35.9 | 17.2 |
| +H,OS,TR | ✓ | ✓ | | ✓ | | 34.0 | 34.5 | 17.0 |
| +H,OS,C | ✓ | ✓ | | | ✓ | 33.8 | 34.2 | 16.9 |
| +H,E,TR | ✓ | | ✓ | ✓ | | 33.5 | 33.8 | 16.7 |
| +H,E,C | ✓ | | ✓ | | ✓ | 33.4 | 33.6 | 16.6 |
| +H,TR,C | ✓ | | | ✓ | ✓ | 33.2 | 33.1 | 16.5 |
| +OS,E,TR | | ✓ | ✓ | ✓ | | 32.9 | 32.8 | 16.4 |
| +OS,E,C | | ✓ | ✓ | | ✓ | 32.8 | 32.7 | 16.3 |
| +OS,TR,C | | ✓ | | ✓ | ✓ | 32.6 | 32.5 | 16.2 |
| +E,TR,C | | | ✓ | ✓ | ✓ | 32.4 | 32.3 | 16.1 |
| +H,OS,E,TR | ✓ | ✓ | ✓ | ✓ | | 35.1 | 37.7 | 17.8 |
| +H,OS,E,C | ✓ | ✓ | ✓ | | ✓ | 34.7 | 37.2 | 17.6 |
| +H,OS,TR,C | ✓ | ✓ | | ✓ | ✓ | 34.2 | 36.5 | 17.3 |
| +H,E,TR,C | ✓ | | ✓ | ✓ | ✓ | 33.9 | 35.7 | 17.1 |
| +OS,E,TR,C | | ✓ | ✓ | ✓ | ✓ | 33.1 | 33.9 | 16.9 |
| **SIMILAR** | ✓ | ✓ | ✓ | ✓ | ✓ | **35.4** | **38.2** | **17.8** |

captures. The *Odds of Success (OS)* dimension follows, with improvements of 31.9%, 24.7%, and 15.4% on the respective benchmarks, indicating its importance in guiding the agent. The *Efficiency (E)* dimension also contributes positively, though to a lesser extent, while *Task Relevance (TR)* and *Coherence (C)* show more modest improvements, consistent with their secondary roles.

### E.2. Combined Impact of Multiple Dimensions

The extended results further highlight the synergistic effects of combining multiple dimensions. For instance, the combination of *H* and *OS* achieves success rates of 33.4%, 31.4%, and 16.7% on Android World, WebArena, and OSWorld, respectively, outperforming models with only one of these dimensions. Similarly, the combination of *H*, *OS*, and *E*

yields even higher success rates (34.3%, 35.9%, and 17.2%), demonstrating the cumulative benefits of integrating complementary dimensions. These results reinforce the importance of fine-grained rewards over coarse-grained ones, as models with partial-dimensional rewards consistently underperform compared to the full `Similar` model.

## F. More Visualizations of *SRMEval*

As depicted in Table 7, we present additional visualizations of *SRMEval*. From the data in the table, we can more clearly understand the content of *SRMEval*, which is the first benchmark in the virtual agent domain designed for step-wise, multi-dimensional, and multi-platform evaluation of reward models. It comprehensively tests the ability of reward models to assess the quality of agent actions, as well as the degree of preference alignment.

Table 7: Cases of *SRMEval*.

| Instruction | Observation | Step Idx | Trajectory | Type | Candidate Action Pair | |
|---|---|---|---|---|---|---|
| In Simple Calendar Pro, delete all the events. |  | 2 | Step 1: Click Search. | E | Click Calender. | Input text "Simple Calendar Pro". |
| In Simple Calendar Pro, delete all the calendar events on 2023-10-27. |  | 6 | Step 1: Click Search. Step 2: Click Calendar. Step 3: Scroll down. Step 4: Long press 27 Friday. Step 5: Click More options. | OS | Navigate back. | Click Import events from an.ics file. |
| Add this exact product to my shopping cart. I think it is in the "Herbs, Spices & Seasonings" category. |  | 3 | Step 1: Click menuitem "Grocery & Gourmet Food" hasPopup: menu. Step 2: Click link "Pantry Staples( 4891 item )". | C | Click link "Herbs, Spices & Seasonings( 707 item )". | Click menuitem "Grocery & Gourmet Food" hasPopup: menu. |
| Add a exact product to my shopping cart. |  | 2 | Step 1: Click menuitem "Grocery & Gourmet Food" hasPopup: menu. | H | Scroll down. | Click menuitem "Grocery & Gourmet Food" hasPopup: menu. |
| Can you add the red flower seeds with around 4 stars to my cart? |  | 4 | Step 1: Scroll down. Step 2: Click link "Page 2". Step 3: Click link "Plants, Seeds & Bulbs( 59 item )" | TR | Click link "Page Next". | Hover menuitem "Patio, Lawn & Garden" hasPopup: menu. |
| Can you make Bing the main search thingy when I look stuff up on the internet? |  | 3 | Step 1: Coordinates for "Customize Chromium". Step 2: Click on "Customize Chromium". | Total | Click on the address and search bar Type the URL to access search engine settings. | Click on the "Account Settings" button coordinates for "Account Settings". |

