# OpenReview forum: "Boosting Virtual Agent Learning and Reasoning: A Step-Wise, Multi-Dimensional, and Generalist Reward Model with Benchmark"
_ICML.cc/2025/Conference — ICML 2025 poster_

### Official Review · Reviewer_Evp4 · 2025-03-11

**Overall Recommendation:** 3

**Summary:**

The paper proposes Similar, a Step-wise, Multi-dimensional Generalist Reward Model, designed to improve the training and inference of Generalist Virtual Agents (GVAs). Similar addresses limitations in outcome-based reward models by introducing a process-based system that provides fine-grained supervision signals. The authors define five dimensions to assess agent actions: Helpfulness, Odds of Success, Efficiency, Task Relevance, and Coherence. They develop an MCTS-P algorithm to automatically annotate agent trajectories across four environments (Web, Android, Linux, and Windows) and train Similar using the Triple-M strategy (multi-step, multi-objective, and multi-modal). Extensive experiments show Similar significantly boosts GVA learning and inference-time scaling.

## update after rebuttal

Thank you to the authors for the detailed response. I am satisfied with the rebuttal and would like to maintain my score after the response.

**Claims And Evidence:**

yes

**Essential References Not Discussed:**

I think that prior works[1,2] also explore process reward models, which should be discussed.

[1] Setlur A, Nagpal C, Fisch A, et al. Rewarding progress: Scaling automated process verifiers for llm reasoning[J]. arXiv preprint arXiv:2410.08146, 2024.
[2] Wang P, Li L, Shao Z, et al. Math-shepherd: Verify and reinforce llms step-by-step without human annotations[J]. arXiv preprint arXiv:2312.08935, 2023.

**Experimental Designs Or Analyses:**

Yes. The experiments are conducted in multiple GVA scenarios with detailed ablations.

**Methods And Evaluation Criteria:**

yes

**Other Comments Or Suggestions:**

Minor: Highlighting relative improvements over the backbone may make them clearer in the tables.

**Other Strengths And Weaknesses:**

Strengths:
--

1. The paper is well-structured and the idea is well-motivated.
2. Implementation details and prompt designs are explained in clear and exhaustive detail.
3. The experimental results are impressive.


Weaknesses:
--

1. I do not understand the significance of the gating network, and there is no ablation study to evaluate it.
2. As mentioned in the manuscript, the proposed method is trained and evaluated on only four benchmarks, making it difficult to assess its scalability to real-world scenarios. Evaluating the trained model on OOD or real-world tasks would be highly beneficial.
3. I am confused about the definition of the Helpfulness metric. Detailed explanations and examples would be greatly appreciated.
4. There is no comparison with the outcome reward model or other works focused on the process reward model, which should be added or the rationale discussed.

**Questions For Authors:**

1. How about the computational costs about the data annotation and training.
2. Could Similar be combined with other inference-time scaling methods, like majority voting and beam search.

**Relation To Broader Scientific Literature:**

The paper connects well to literature on reward models, GVA training, and MCTS-based data collection.

**Theoretical Claims:**

There is not theoretical claim in this submission.

---

> ### Author Rebuttal · Authors · 2025-03-31
>
> **Q1: Compared with more process reward models (PRMs) and outcome reward models (ORMs).**
> **A1:** Thank you for suggestions on more comparisons with more RMs, To our knowledge, our **Similar** is the first step-wise, multi-dimensional, cross-platform PRM for *virtual agents (VA)*. **PAVs[1]** is a mathematical reasoning PRM that uses advantages as a single-dimensional reward. **Math-shepherd[2]** is a PRM for mathematical reasoning that uses soft estimation as a single-dimensional reward. Through experimentation, we found **these RMs are not applicable** as RMs for our proposed multi-dimensional, cross-platform VA tasks.
>
> Following your nice advice, we try to apply these RMs in VA tasks in following comparative experiments:
> |RM|Android World(AW) SR (%)|WebArena(WA) SR (%)|
> |-|-|-|
> |General ORM[3]|31.0|21.4|
> |Math-Shepherd[2]|31.5|22.9|
> |PAVs[1]|31.3|23.8|
> |**Similar**|**35.4**|**38.2**|
>
> The table shows that our model outperforms other RMs, likely due to its multi-dimensional scoring designed for VA tasks. In contrast, these RMs only provide either outcome rewards or single-dimensional process rewards, making them unsuitable for VA tasks.
>
> [1] Setlur A, et al. Rewarding progress: Scaling automated process verifiers for llm reasoning. 2024.
>
> [2] Wang P, et al. Math-shepherd: Verify and reinforce llms step-by-step without human annotations. 2023.
>
> [3] Hunter L, et al. Let's Verify Step by Step. 2023.
>
> **Q2: Significance of gating network.**
> **A2:** The gating network (GN) **balances the 5 dimensions, addressing multi-objective optimization challenges**. For example, *Task Relevance* may dominate in instruction-following tasks, while *Efficiency* matters more in time-sensitive scenarios. The GN is not the focus of our research, but we will consider it in the future. The ablation experiments we have added show that removing the GN degrades performance, validating its necessity:
> - **Similar (w/o GN)**: AW SR = 33.7%, WA SR = 34.1%
> - **Similar (with GN)**: AW SR = **35.4%**, WA SR = **38.2%**
>
> **Q3: 4 benchmarks and more OOD experiments.**
> **A3:** Thank you for your suggestions on more OOD experiments. 4 benchmarks (WebArena, VisualWebArena, Android World, OSWorld) mentioned in our paper are **widely recognized** and cover the major **real-world online interactive environments** of Web, Android, Linux, and Windows, as well as real-world scenarios such as web navigation and app usage. Thus, we believe that these benchmarks can already reflect the situation of real scenes.
>
> Further, as you nicely suggested, we supplement 3 **OOD tests**: Game of 24 (a mathematical puzzle unrelated to virtual agents), ScienceWorld (embodied science experiments), and 30 real-world Android tasks we designed (e.g., ordering bubble tea). Experimental results are shown below:
> |Benchmark|Platform|Baseline SR (%)|Similar SR (%)|
> |-|-|-|-|
> |Game of 24|Virtual Game|1.0|**3.0**|
> |ScienceWorld[4]|Embodied Science|unseen: 18.7|**unseen: 20.1**|
> |Real Android Device (designed by ourselves)|Physical Phone|16.7|**46.7**|
>
> These results confirm our model’s scalability beyond the original benchmarks. We will include full details in the next version.
>
> [4] Ruoyao W, et al. ScienceWorld: Is your Agent Smarter than a 5th Grader? 2022.
>
> **Q4: *Helpfulness*.**
> **A4:** We are sorry for not clearly explaining this dimension's definition. The ***Helpfulness* quantifies each step's contribution to task completion** by assigning values inversely proportional to the trajectory length. For example, each step in a 3-step successful trajectory is worth 1/3, while steps hindering progress (those failing to lead to success) receive corresponding negative values. To account for the compensatory relationship between helpful and harmful steps, we introduce an intermediate accumulator (AC) to track each step's cumulative influence throughout the trajectory.
>
> **Q5: Other inference-time scaling methods.**
> **A5:** Our model can seamlessly integrate with methods like majority voting and beam search. For example, we apply beam search as follows:  1) At each step, Similar scores k=5 candidate actions. 2) The top k=2 paths are expanded further, guided by our rewards. Additional experiments demonstrate this flexibility:
>
> - **Similar + Beam Search (k=5)**: AW SR = 34.1%, WA SR = 36.1%
> - **Similar + Majority Voting (N=10)**: AW SR = 34.3%, WA SR = 35.6%
>
> **Q6: Computational costs in data annotation and training.**
> **A6:**
>
> **Annotation**: The 5 dimensions can directly derive from a single MCTS process. While MCTS introduces computational costs, we reduced them via pruning and limiting the number of rollouts (e.g., N=8).
> - Generating 10k trajectories required ~15,000 GPT-4o API calls, and the total annotation time is ~80 hours
> - Parallel MCTS rollouts with pruning reduced per-task latency by ~30%
>
> **Training**: The model is trained on preference pairs using lightweight regression with Bradley-Terry loss.
> - Training Similar on 78k preference pairs took ~100 GPU hours

---

### Official Review · Reviewer_d34U · 2025-03-13

**Overall Recommendation:** 3

**Summary:**

Previous multimodal LLM-based virtual agents usually require human annotations, multi-dimensional fine-grained process supervision, and scaling inference time. The paper proposes a new step-wise, multi-dimensional generalist reward model to offer fine-grained signals for agent training and can choose better actions for inference-time scaling. Specifically, the paper first provides five dimensions for agent action evaluation, including helpfulness, odds of success, efficiency, task relevance, and coherence. The first three dimensions can be computed automatically, and the other two can be evaluated by MLLMs.  The paper then designs a Monte Carlo Tree Search-Plus to collect excutation data. Finally, the paper employs a multi-step, multi-objective, and multi-modal strategy to train a reward model. Moreover, the paper proposes a new virtual agent benchmark for step-wise, multi-dimensional reward model training and evaluation. The SRM benchmark is based on GPT-4o-1120's agent action on WebArena (WA), VisualWebArena (VWA), Android World (AW), and OSWorld (OW). The paper selects 32k data for manual annotation and as a test set. The paper also proposed a new task focusing on the candidate action of the MLLM agent. The paper tests two baseline models, Qwen2VL and Llama-3.2-Vision-Instruct. The paper analyzes the alignment of preference, RL-training using the proposed dataset, and Inference-time scaling. The paper also includes an ablation study, a case study, and a correlation study.

**Claims And Evidence:**

1. The paper claims that Triple-M can provide robust fine-grained feedback for the model. Tables 1, 2, and 3 show that the baseline adding triple-M can improve the performance.
2. The paper also introduces SRM-eval and SRM-train. However, in the experiment, the paper fails to show the benefit of using SRV-eval instead of traditional metrics in section 4. Including similar benchmarks for comparison will strengthen the argument by demonstrating that previous benchmarks are insufficient for fully evaluating the model's capabilities.

**Essential References Not Discussed:**

N/A

**Experimental Designs Or Analyses:**

The paper conducts relatively comprehensive experiments to measure the performance of the proposed method. The paper evaluates the alignment preference in the training stage to analyze the effectiveness of the proposed dataset and training method. However, the whole paragraph seems to only draw conclusions from Table 1, with two findings supported by Table statistics. Adding additional qualitative analysis can strengthen this paragraph. Similarly, for section 5.3, the authors can remove the contents already shown in the paper and move some of the analysis to the appendix here.

**Methods And Evaluation Criteria:**

1. The proposed five dimensions for step-wise assessment are not clear. What is the AC in line 207? What is the M here? The paper seems to use human evaluation to judge the quality of the proposed assessment data. However, none of the human evaluation details are provided in Appendix C. Additionally, why are the sample sizes for human evaluation not the same? Given that the dataset SRM is constructed with GPT-4o, why does the knowledge distillation model,  Llama 3.2-V + SimilarRL, perform better than the original GPT-4o?
2. The paper uses Triple-M Strategy for Reward Model Training. What is the pretrained decoder-only MLLM uised here? is it the backbone of training data?

**Other Comments Or Suggestions:**

N/A

**Other Strengths And Weaknesses:**

Pros:
* The visualization, examples, and tables are clear to readers and easy to understand.

Cons:
* The paper is not well written and contains a lot of abbreviations without proper definitions. The abstract contains too many abbreviations, such as MCTS-P and Triple-M, which are not defined until the introduction section. The MCTS-P is really confusing since only the full name of MCTS exists in the paper. The paper should not assume all the readers will check the paper carefully. Additionally, the paper sometimes uses Triple-M and sometimes uses X+3M, which is very confusing.

**Questions For Authors:**

See above

**Relation To Broader Scientific Literature:**

The paper proposes a new interesting multi-dimensional model with a five-dimensional reward to provide better feedback. The paper also adopts the multi-step, multi-objective, and multi-modal to train the model. The paper finally release a new dataset called SRM.

**Theoretical Claims:**

N/A

---

> ### Author Rebuttal · Authors · 2025-03-31
>
> **Q1: Benefit of *SRMEval* as a benchmark.**
> **A1:** Thank you for suggestions on more comparisons with other benchmarks. Section 1 of our paper proposes that ***SRMEval*** is **the first multi-step, multi-dimensional, and multi-platform benchmark** specifically for **evaluating reward models(RMs) in virtual agents**, featuring 32k professionally annotated preference pairs. Our Section 5.2 experiments further demonstrate its challenging nature. We summarize and compare the benefits of our RM:
>
> |Benchmark|Target|Step-wise|Multi-Dimension|Platform|Size|
> |-|-|-|-|-|-|
> |RewardBench[1]|General RM|✗|✗|N/A|3k|
> |RMB[2]|General RM|✗|✗|N/A|18k|
> |WebArena| Virtual Agent|✗|✗|Web|812|
> |OSWorld|Virtual Agent|✗|✗|Linux, Windows|369|
> |***SRMEval***|**RM for Virtual Agent**|**✓**|**✓**|**Web, Android, Linux, Windows**|**32k**|
>
> Through experimentation, we find these benchmarks cannot evaluate RMs for Virtual Agent, which constitutes *SRMEval*'s benefit. We find that **high-performing RMs on *SRMEval* also boost agent performance across environments**, which also indicates the benefit of *SRMEval*. Relevant experiments are in our reply to Reviewer **sojt's Q4**.
>
> [1] Nathan L, et al. RewardBench: Evaluating Reward Models for Language Modeling. 2024.
>
> [2] Enyu Z, et al. RMB: Comprehensively Benchmarking Reward Models in LLM Alignment. 2024.
>
> **Q2: AC and M.**
> **A2:** We are sorry for not clearly explaining the definition of our dimensions. And we appreciate the opportunity to clarify these terms:
> - **$ AC $**:  Purely an intermediate computational variable with no standalone semantic meaning. It serves as a mathematical placeholder to recursively track cumulative *Helpfulness* scores during MCTS rollouts.
> - **$M$**: Already defined in Section 3.1 under *Helpfulness*:  "*M* is the total number of reasoning steps".
>
> For a detailed explanation of *Helpfulness*, please refer to our reply to Reviewer **Evp4's Q4**.
>
> **Q3: Human evaluation details.**
> **A3:** Thank you for suggestions on more human evaluation details. The **human evaluation details** on how to train annotators and ensure data quality can be found in our response to Reviewer **sojt's Q4**.
>
> The **Human Acceptance in Appendix C** is that, we randomly sampled data from *SRM* and asked trained annotators to verify these data. If the annotator considers a sample is correct, mark it as 1; otherwise, mark it as 0, and calculate Accuracy as Human Acceptance.
>
> **Q4: Sample size variation.**
> **A4:** The varying sample sizes occur because the **five dimensions inherently contain different amounts of data**. This stems from their fundamental design - for instance, *Task Relevance* and *Coherence* are binary values, which naturally yield fewer possible preference pairs.
>
> The distribution of dimensions in *SRM* is roughly as follows:
> |Dimension|Size|
> |-|-|
> |Helpfulness|25k|
> |Odds of Success|15k|
> |Efficiency|25k|
> |Task Relevance|5k|
> |Coherence|10k|
> |Total|10k|
> |Traj|20k|
>
> We think dimensions with small amounts of data indicate weak discriminability and are easily trained for prediction.
>
> **Q5: Why is our model better than GPT-4o in *SRMEval*?**
> **A5:** We appreciate the opportunity to clarify "why our model performs better than GPT-4o":
>
> - **GPT-4o** was only used to provide reliable labels for binary *Task Relevance* and *Coherence* dimensions in SRM construction. For *SRMEval* testing, **GPT-4o served as a zero-shot reward model**. It failed to accurately evaluate the three formula-based dimensions (*Helpfulness/Odds of Success/Efficiency*), causing unreliable predictions.
> - Conversely, our **dedicated reward model (Llama 3.2-V + SimilarRL)** was trained on 78k multi-dimensional preference pairs using our Triple-M strategy. The model employs a purpose-built architecture with gating and regression layers for precise prediction of the benchmark's action pairs.
>
> **Q6: Pretrained MLLM and backbone.**
> **A6:** We used **Qwen2-VL-7B-Instruct** and **Llama-3.2-11B-Vision-Instruct** as frozen backbones (mentioned in Section 5.1) which are also the pretrained MLLMs.
>
> **Q7: Lack of qualitative analysis?**
> **A7:** We appreciate this suggestion. But in fact, we have provided qualitative analyses in: **1) Fig. 5**: Illustrates how Similar guides the agent during both the training and testing phases. **2) Appendix D**: Includes 6 visible detailed cases.
>
> As you nicely suggested, we add textual descriptions of **failure cases qualitative analysis**, which we will continue to explore and improve our dimension weighting strategy in the future:
> - ***Efficiency* Misjudgment**: Prioritized shortcut clicks over explicit menu navigation, even when shortcuts were contextually invalid.
> - ***Coherence* Over-penalizing**: In a "file backup" task, our model incorrectly penalized opening a text editor and file explorer simultaneously, though both were necessary.
>
> **Q8: Writing issues.**
> **A8:** We thank you for the suggestions and will address these writing issues in the next version.

---

> > ### Comment · Reviewer_d34U · 2025-04-02
> >
> > Thank you for your comments. The authors have addressed all my questions. Therefore, I decided to raise my score to 3. Unfortunately, I cannot see the updated version of the paper.

---

> > > ### Author Response · Authors · 2025-04-02
> > >
> > > We sincerely appreciate your consideration in revising the score after reviewing our response—this is a significant recognition of our work. We truly hope our clarifications have adequately addressed your concerns.
> > >
> > > Regarding your note about not seeing an updated paper version, we regret that ICML 2025’s policy explicitly states in the *Author Instructions* under *Reviewing and Author Response* that ***"There is no option to upload a revised version of the paper during the author feedback period."*** We will however carefully incorporate all constructive reviewers' feedback in the paper’s next version.
> > >
> > > Thank you again for your thorough and insightful review, and for acknowledging our paper’s merits.

---

### Official Review · Reviewer_sojt · 2025-03-14

**Overall Recommendation:** 3

**Summary:**

Traditional training methods for virtual agents depend on outcome supervision and costly human annotations, limiting their scalability. The authors propose Similar, a step-wise multi-dimensional reward model that refines agent training and improves inference-time decision-making. They define five key dimensions for evaluating agent actions and introduce MCTS-P, an automated algorithm for collecting and labeling execution data. Using this data, Similar is trained with the Triple-M strategy and the authors also introduce SRM, the first benchmark designed for step-wise, multi-dimensional reward model training and evaluation, consisting of SRMTrain for training and SRMEval for testing. Experiments demonstrate that Similar provides more granular feedback, enhancing both training efficiency and real-time decision-making for virtual agents.

**Claims And Evidence:**

From the performance perspective, the argument is well-supported by their proposed process reward modeling methods. However, weaknesses remain in the experimental design and the fairness of evaluation settings, which will be discussed in the following sections.

**Essential References Not Discussed:**

N/A

**Experimental Designs Or Analyses:**

While the research direction is promising with significant potential impact, several weaknesses in the methodology and experimental design should be addressed:

**Claim on no manual annotations**: The paper asserts that the method requires no manual annotations. However, pruning trajectories and verifying correctness depend on human-annotated evaluation scripts from (Visual)WebArena, AndroidWorld, and OSWorld, which involve substantial human effort. This contradicts the core claim that the process is fully automated.

**Evaluation in Tables 2 and 3**: The training data appears to be sampled from WebArena, AndroidWorld, and OSWorld, which are benchmarks originally designed for testing. This raises concerns about potential overlap between SRMEval’s training data and downstream evaluation tasks. The authors should clarify the sampling process to ensure fair evaluation.

**Total number of trajectories**: Given that (Visual)WebArena, AndroidWorld, and OSWorld contain only a few hundred tasks each, it is unclear how the authors sampled 10K trajectories from these benchmarks. Further explanation is needed to validate the dataset scale.

**Reliability of the test set**: Although human annotators create the test set, their accuracy remains uncertain. Determining the correct UI interactions can be challenging, especially for unfamiliar websites and apps. A more detailed discussion on monitoring annotation quality and potential errors would strengthen the evaluation.

**Methods And Evaluation Criteria:**

The proposed methods and evaluation benchmarks are well-designed, achieving top performance on both the in-house annotated SRMEval and widely used datasets such as OSWorld and WebArena.

**Other Comments Or Suggestions:**

N/A

**Other Strengths And Weaknesses:**

N/A

**Questions For Authors:**

**Effect of the five reward dimensions**: It would be insightful to analyze whether some reward dimensions influence performance more than others. For instance, should helpfulness weigh more than coherence? Instead of a pure sum, a weighted sum approach could better reflect their relative impact.

**Relation To Broader Scientific Literature:**

Compared to previous works, this paper introduces a novel perspective on process reward modeling by incorporating multiple dimensions, which involves direct and meaningful connection to actual agent performance.

**Theoretical Claims:**

N/A

---

> ### Author Rebuttal · Authors · 2025-03-31
>
> **Q1: Claim on no manual annotations.**
> **A1:** We appreciate to **clarify a misunderstanding**: our claim of "no manual annotations" means while utilizing the benchmarks' evaluation scripts (for dimension calculations), our method **annotates step-wise multi-dimensional data without any additional human effort**. **1)** In fact, these scripts are all readily available and involve only simple result-based judgments - neither complex nor labor-intensive, just basic API calls. The human effort required to write these scripts is very limited. Notably, nearly all related works use these scripts. **2)** In contrast, manually annotating the 110k step-wise multi-dimensional data provided by our automated annotation framework would require substantial human effort. Our experiments show even trained annotators can only process ~**25 step-wise annotations/hour**.
>
> Further, we invited researchers to reproduce these evaluation scripts. We compared their completion time against our estimated time for manually annotating our 110k data:
>
> |Work|Task|Size|Labor Effort|
> |-|-|-|-|
> |WebArena|Scripts Coding|N/A|2.5h|
> |AndroidWorld|Scripts Coding|N/A|2h|
> |OSWorld|Scripts Coding|N/A|4h|
> |**Similar (w/o autonomous annotation)**|Step-wise Multi-dimensional Data Annotation|110k|**4400h**|
> |**Similar**|Step-wise Multi-dimensional Data Annotation|110k|**0h**|
>
> The table shows the coding time for evaluation scripts is negligible (relative proportion **0.19%**) compared to the manual time saved by our approach.
>
> **Q2: Evaluation in Tables 2 and 3.**
> **A2:** We rigorously ensured **no data overlap** between training (*SRMTrain*) and evaluation (4 benchmarks) sets by randomly splitting the original test tasks from WebArena(WA), AndroidWorld(AW), and OSWorld into **30% for our training data collection** and **70% for evaluation** (as detailed in Section 4).
>
> We further validate the fairness of our configuration through **OOD experiments** (in our reply to Reviewer **Evp4's Q3**), demonstrating our model's robust performance on out-of-distribution data.
>
> **Q3: Total number of trajectories.**
> **A3:** Thank you for your question. We achieved 10k trajectories sampling by **generating multiple distinct actions per step** through task-specific prompt injection and stochastic exploration. For a 5-step task with 5 action variations per step, this theoretically yields $5^5=3125$ unique trajectories (before deduplication). In this way, we easily generated 10k trajectories.
>
> **Q4: Reliability of the test set.**
> **A4:** To ensure high-quality annotations, we collaborated with a professional commercial data labeling team. The process included: **1) Training Phase**: Annotators underwent three rounds of iterative "label-review-feedback" cycles to clarify ambiguities of annotation (e.g., the complexity of UI interaction tasks). Only after achieving >95% accuracy on validation samples did formal annotation begin. **2) Formal Annotation**: Each test sample in *SRMEval* was independently labeled by three annotators and three checkers. The final data in test set required >99% accuracy.
>
> Besides, we find that high-performing RMs on *SRMEval* also boost agent performance across environments. This finding additionally validates the reliability of the test set (*SRMEval*), motivating the following comparative experiments:
>
> |RMs|SRMEval Acc (%)|SR Improvement on AW (%)|SR Improvement on WA (%)|
> |-|-|-|-|
> |Qwen2-VL|10.3|0.4|0.6|
> |Qwen2-VL + Similar-RL|25.7|1.7|5.7|
> |Qwen2-VL + Similar-3M|46.5|3.8|14.3|
>
> As shown in the table, achieving better performance on *SRMEval* often leads to greater improvement in agent performance.
>
> **Q5: Effect of the five reward dimensions.**
> **A5:** Thank you for your suggestions regarding the 5-dimension weighting scheme. As shown in Tab. 4, Section 5.5 and Appendix E's ablation study, we found the 5 dimensions' relative importance follows approximately H > OS > E > TR > C. We find these dimensions show similar importance patterns in healthcare, education, and other fields[1].
>
> Aligning with your insightful observation, we actually avoided using equal weights and instead **adopted the weighted scoring scheme** presented in Section 3.2: **$5×H + 3×OS + 3×E + TR + C$**, with weights based on empirical analysis. As you nicely suggested, we conducted additional experiments comparing different weighting schemes:
>
> |Method|Android World SR (%)|WebArena SR (%)|
> |-|-|-|
> |Similar + Equal Weights|33.9|35.3|
> |Similar + Dynamic Weights[2]|34.8|**38.4**|
> |**Similar (Fixed Weights)**|**35.4**|38.2|
>
> The table demonstrates that the dynamic weights strategy also achieves significant improvements, we will explore optimal weighting in future work.
>
> [1] Hattie J. Visible Learning: A Synthesis of Over 800 Meta-Analyses Relating to Achievement. 2008.
>
> [2] Aminul H, et al. Adaptive Weight Assignment Scheme For Multi-task Learning. 2023.

---

### Decision · Program_Chairs · 2025-05-01

**Decision:**

Accept (poster)

**Comment:**

This paper proposes a method for training multimodal virtual agents, using reward models around 5 dimensions for training and evaluation. The approach is unique and somewhat timely given the difficulty of both training and evaluating virtual agents, but more importantly permits using LLMs in various ways to provide a variety of signal to the virtual agent. The benchmarks are sound and cover both well-known problems as well as OOD generalization.

There were various concerns about clarity which have been well resolved in the discussion. I generally found the questions from reviewers to be unconfused, meaning the core ideas of the paper were clear and there were no significant issues on support for the central message. All in all, I found the responses to be sufficient to address a large majority of the concerns.

I therefore recommend the paper is accepted to the conference.